

# Further aspects of Supersymmetric Virasoro Minimal Strings

Clifford V. Johnson*

*Department of Physics, Broida Hall, University of California, Santa Barbara, CA 93106, U.S.A.*

A supersymmetric version of the Virasoro minimal string was defined some time ago using random matrix model techniques. Several of the special properties of the matrix model that were noted (accessible fully non-perturbatively) underlie features shared by both 0A and 0B versions of the theory. This paper develops the formalism much further, including showing how the 0A and 0B choices familiar in continuum approaches have a direct analogue in terms of building solutions of the appropriate string equations. This also illuminates several key differences between the 0A and 0B models at the level of matrix model loop observables. The natural all-orders vanishing of loops, already observed in some 0A models, translates into the same for 0B. It is also noted that the leading amplitude for a single asymptotic boundary, as well as the trumpet partition function, are characters of a 2D superconformal field theory living on the boundary of a solid torus, suggesting a 3D chiral supergravity dual. Non-perturbative results are computed as well.

## I. INTRODUCTION

Double-scaled random matrix models [1–4] are excellent tools for formulating certain classes of string theory, and more generally, various 2D gravity theories. As models of ensembles of $N \times N$ matrices, they have two complementary points of entry in this gravity context. On the one hand, they are statistical characterizations of universal properties of Hamiltonians of systems that are holographically dual to the 2D gravity theory of interest, while on the other hand they are tools for efficiently regulating the sum over geometries and topologies needed to formulate the quantum gravity problem as a path integral. These two points of view (characterized as *Wignerian* and *'t Hooftian* in an essay on the two interpretations [5]) come together rather successfully in the study of the near horizon dynamics of near-extremal black holes, where the effective model is Jackiw-Teitelboim (JT) gravity [6, 7].

Double-scaling refers to the process of taking $N$ large while also tuning into the universal physics to be found in the vicinity of a critical point. This washes away the non-universal details of the descretization of the worldsheet that the matrix model provides (through 't Hooftian ribbon diagrammatics) [8–10], capturing a sensible continuum limit that makes contact with other approaches.[1]

In the string context, random matrix models were shown to capture theories where the world-sheet sector is comprised of a (spacelike) Liouville conformal field theory (CFT) (the "gravity" sector) coupled to an ordinary "matter" CFT. The central charges of the two sectors add

to the total required to cancel the conformal anomaly and produce a consistent string theory.

Recently, however, ref. [16] introduced and explored a new class of string theory called the "Virasoro minimal string" (VMS). Instead of the traditional matter CFT sector, the spacelike Liouville theory is joined by a timelike Liouville CFT. The spacelike Liouville central charge is $c = 1 + 6Q^2 \geq 25$, with $Q = b + b^{-1}$ and $0 \leq b \leq 1$. The timelike Liouville has the corresponding quantities (denoted with hats), $\hat{c} = 1 - 6\hat{Q}^2 \leq 1$, $\hat{Q} = b^{-1} - b$ such that $c + \hat{c} = 26$. Also, $\hat{P} = iP$, and the Virasoro weights of each sector are $h_P = \frac{Q^2}{4} + P^2$, $\hat{h}_{\hat{P}} = -\frac{\hat{Q}^2}{4} + \hat{P}^2$, so that $h_P + \hat{h}_{\hat{P}} = 1$. One of the remarkable aspects of the theory is that its main observables (correlators of $n$-point insertions of some Liouville momenta $P_i$), which have a (genus) perturbative expansion in terms of the topology of the worldsheet, are geometrical objects that can be computed from several different perspectives. These "quantum volumes" $\hat{V}_{g,n}(\{P_n\})$ can be computed as stringy scattering amplitudes in the 2D gravity (Liouville) theory, but they also have a description in a 3D chiral gravity model [17] defined on the solid torus $\Sigma_{g,n} \times S^1$ (where $\Sigma_{g,n}$ is a 2D hyperbolic surface with $g$ handles and $n$ boundaries). There, they are partition functions in the (AdS$_3$/CFT$_2$) holographically dual CFT. In the classical limit $b \to 0$, the 2D gravity theory reduces to JT gravity, and the quantum volumes become the Weil-Peterson volumes $V_{g,n}$, the volume of the moduli space of hyperbolic Riemann surfaces of genus $g$ with $n$ geodesic boundaries. The momentum insertions $\{P_i\}$ become the lengths $\{\ell_i\}$ of the geodesic boundaries in the limit. This interplay of descriptions has been very instructive, providing a new laboratory for studying holography, as well as non-trivial string theory backgrounds.

A third, very powerful setting for the VMS is the double-scaled random matrix model description that ref. [16] developed. A key (and remarkable) point is that

---

* cliffordjohnson@ucsb.edu

[1] This is reviewed in *e.g.* ref. [11] for string world-sheet formulations. The applications to new (Jackiw-Teitelbiom-like) classes of gravity theory began in refs. [12, 13], and reviews can be found in *e.g.* refs. [14, 15]

the expectation value of a loop of fixed length $\beta$ in the matrix model computes (the $S$-transform of) the vacuum partition function of the CFT$_2$ that is holographically dual to the aforementioned 3D chiral gravity on the solid torus. This and other aspects of the random matrix model will be reviewed shortly in Section I A. A random matrix model gives an efficient way of computing the quantum volume observables (after dressing with asymptotic boundaries they are loop correlators in this context) to all orders in perturbation theory, and moreover give the most direct access (when available) to information beyond perturbation theory. Indeed, when formulated as a family of multicritical models (as done in refs. [18, 19]) techniques such as orthogonal polynomials, string equations, an underlying KdV integrable system, and more, can be used to efficiently extract a great deal of perturbative knowledge about the VMS, as well as diagnose non-perturbative content.

It is the random matrix model approach that will be the main focus of this Paper. Ref. [20] introduced a random matrix model that naturally describes (fully non-perturbatively) an $\mathcal{N}{=}1$ supersymmetric generalization of the Virasoro minimal string (SVMS) that has many rather special properties, including being fully non-perturbatively well-defined. The basic points underlying the construction are introduced shortly in Section I B, and expanded upon later in the paper. As part of the motivation, Section I B also uncovers how the random matrix model partition function is a natural vacuum partition function in a superconformal CFT$_2$ that is almost certainly dual to 3D chiral supergravity.

The purpose of this paper is to explore further the properties of the (type 0A) random matrix model defining the SVMS and also show how to use its components to define a 0B variant of the model that has several special properties. Many of the results obtained in this paper (including the specific construction that takes solutions of a string equation and builds either 0A to 0B string theories) are more generally applicable than just to SVMS models. Section I C contains a detailed summary of many of the paper's results.

Since ref. [20]'s definition of the supersymmetric Virasoro minimal string using matrix model techniques (in the absence of any definition at the time), two heroic sets of computations using super-Liouville (timelike and spacelike) techniques have entered the fray [21, 22], beginning the task of formulating the CFT elements that could yield SVMS using a continuum approach. Such work (and more of it) is of course needed, as a complement to other approaches (they are also needed for better understanding of timelike Liouville). Indeed, the supersymmetric 3D chiral gravity approach strongly suggested by the observations of Section I B should be pursued as well.

So far, however, the only fully operational definition of the SVMS is the matrix model described here and in ref. [20]. After 30 or more years of successes for these techniques in this arena, it should not need to be stated

that a matrix model approach to formulating new kinds of string theory is a valuable and an especially powerful part of the toolbox of approaches for learning about string theory. It is hoped that the results presented here serve as a useful guide to other approaches in this exciting new frontier.

## A. VMS Background Remarks

Ref. [16] showed that the VMS is perturbatively (in worldsheet topology) equivalent to a random matrix model with a leading spectral density $\rho_0^{(b)}(E)$ that is the universal Cardy density of states in a generic 2D CFT:

$$\rho_0^{(b)}(E) = 2\sqrt{2}\frac{\sinh(2\pi b\sqrt{E})\sinh(2\pi b^{-1}\sqrt{E})}{\hbar\sqrt{E}} \ , \qquad (1)$$

where energy $E$ is related to the Liouville momentum by $E{=}P^2$, and $\hbar{\equiv}e^{-S_0}$ emerges from the matrix model large $N$ limit as a (renormalized) $1/N$ parameter.[2] The continuous parameter $b$ is defined in the opening paragraphs above.

That a random matrix model with spectral density (1) emerges as a description is, in some regards, a bit mysterious. For a start, which CFT is now being discussed, and how does the matrix model relate to it? The natural setting, explained in ref. [16] helps motivate things,[3] building on previous work in ref. [23]. The context is 3D chiral gravity on the solid torus $D^2{\times}S^1$, where $D^2$ is the hyperbolic disc with (renormalized) boundary length $\beta$. Holography suggests a description in terms of a CFT$_2$ on the boundary of the torus. See figure 1.

Laplace transforming the density (1) gives the expectation value, denoted $Z_0$, of a loop of length $\beta$ in the matrix model (the object that reduces to the JT gravity disc partition function in the classical limit). It is:

$$\hbar Z_0 = \sqrt{\frac{2\pi}{\beta}}\left(e^{\frac{\pi^2 Q^2}{\beta}} - e^{\frac{\pi^2 \widehat{Q}^2}{\beta}}\right) = \sqrt{\frac{2\pi}{\beta}}q^{-\frac{(c-1)}{24}}\left(1 - q\right) \ , \tag{2}$$

with $q{\equiv}e^{-\frac{4\pi^2}{\beta}}$. The first form has a striking symmetric separation between the $Q$ and $\widehat{Q}$ (spacelike and timelike Liouville) sectors. The second form writes it all in terms of the central charge $c$ of the spacelike Liouville system. This form has a nice interpretation in terms of the CFT$_2$ on the torus. Notice that the torus in question is rectangular and so the modulus $\tau{=}\tau_1{+}i\tau_2$ has $\tau_1{=}0$. In the thermal presentation of the CFT, the period of

————

[2] In the classical limit $b{\to}0$ where the system becomes JT gravity, which appears in the near-horizon limit of certain low temperature black holes, $S_0$ is the extremal entropy of the black holes. Perturbation theory in $\hbar$ is then working in the semi-classical "large black hole" regime.

[3] The Author thanks Victor Rodriguez for a helpful conversation about this.

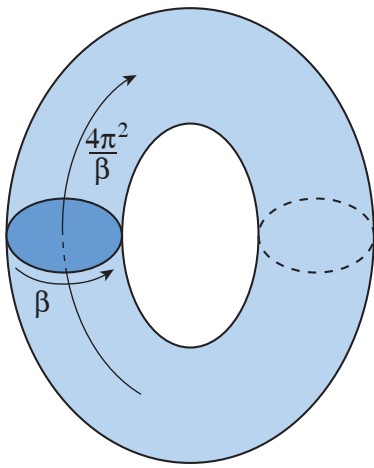

Figure 1. The solid torus $D^2 \times S^1$, with the two dual cycle periods indicated.

the circle sets the inverse temperature: $1/T = 2\pi\tau_2$ and here it is $4\pi^2/\beta$, *i.e.*, going inversely with the length of the loop in the matrix model. They are related by the $S$-transformation: $\tau \to -1/\tau$, yielding the dual channel with $\widetilde{q} = e^{-\beta}$. With this all in mind, the partition function (2) is precisely the $S$-transformed holomorphic piece of a simple CFT vacuum partition function (or character):

$$Z(\tau) = \widetilde{q}^{-\frac{c}{24}} \prod_{n=2}^{\infty} (1 - \widetilde{q}^n)^{-1} = \widetilde{q}^{-\frac{c-1}{24}} \frac{(1 - \widetilde{q})}{\eta(\widetilde{q})} , \qquad (3)$$

where $\eta(\widetilde{q}) \equiv \widetilde{q}^{-\frac{1}{24}} \prod_{n=1}^{\infty} (1 - q^n))$, but with a factor of $\eta^{-1}(\widetilde{q})$ stripped off. (The extra factor in (2) comes from the fact that $\eta(-1/\tau) = \eta(\tau)\tau_2^{\frac{1}{2}}$.) As explained in refs. [16] (see also refs.[23–25]), the absence of that factor is natural since the contribution of Virasoro descendants is not statistically independent of the primaries, and hence the matrix model determines features of the latter and not the former.

There is a similar story for another important object, the "trumpet" partition function, which connects a geodesic boundary associated with momentum $P$ with an asymptotic boundary of length $\beta$. For example, the genus $g$ correction to the amplitude just discussed, $Z_g(\beta)$ is computed using the trumpet as follows:

$$Z_g(\beta) = \int_0^{\infty} 2P dP \; Z_{\mathrm{tr}}(\beta, P)\widehat{V}_{g,1}(P) ,$$

$$\text{where} \quad Z_{\mathrm{tr}}(\beta, P) = \sqrt{\frac{2\pi}{\beta}} e^{-\frac{4\pi^2 P^2}{\beta}} = \sqrt{\frac{2\pi}{\beta}} q^{P^2} , \quad (4)$$

using the conventions of ref. [16], and $\widehat{V}_{g,1}$ are the quantum volumes associated to the surface with $g$ handles and one geodesic boundary. The trumpet, through this gluing integral and with this normalization, is the fundamental object that connects the geodesic holes on any $\widehat{V}_{g,n}(\{P_i\})$ to $n$ asymptotic boundaries with lengths $\{\beta_i\}$.

Its CFT$_2$ interpretation, suggested by the second form in (4), is simply the partition function of a state with weight $h_P = \frac{c-1}{24} + P^2$:

$$\chi_{h_P}(\tau) = \widetilde{q}^{h_P - \frac{c}{24}} \prod_{n=1}^{\infty} (1 - \widetilde{q}^n)^{-1} = \frac{\widetilde{q}^{P^2}}{\eta(\widetilde{q})} , \qquad (5)$$

but $S$-transformed to the channel with $q = e^{-\frac{4\pi^2}{\beta}}$ and with the $\eta$-function again stripped off. So, remarkably, the Liouville momentum $P$ is then naturally in holographic correspondence with states propagating in the $S$-dual channel associated with the $S^1$ direction with period $4\pi^2/\beta$.

In the next (sub)Section), a very natural extension of these results for the $\mathcal{N}=1$ supersymmetric case will emerge. The existence of a SCFT$_2$ will strongly suggest a dual 3D chiral supergravity.

Before proceeding, it is worth recalling the work of refs. [18, 19], in which the VMS was formulated as a multicritical matrix model. There is a useful basis of simple models indexed by $k$, labelling a characteristic $E^{k-\frac{1}{2}}$ behaviour in their spectral density. A general matrix model can be built by combining all of them with relative admixture given by constants $t_k$. (Some of this terminology will be reviewed in Section II.) The formula is:

$$t_k = 2\sqrt{2}\pi \frac{\pi^{2k}}{(k!)^2} \left( Q^{2k} - \widehat{Q}^{2k} \right) . \qquad (6)$$

These $t_k$ serve to encode the leading spectral density into the non-linear ordinary differential equations (ODEs) or "string equations" defining the matrix model. Henceforth, $\hbar$ corrections can be determined perturbatively through recursive solution of the ODE, and (where available) non-perturbative information as well.

Notice that the clean separation between the $Q$ and $\widehat{Q}$ sectors happens in (6) as well. Put differently, the spacelike and timelike Liouville sectors are built as two separate (isomorphic) towers of multicritical models that are then added together, which is interesting. An important special case is when $b=1$, for then $\widehat{Q}=0$, and the $t_k$ formula becomes the multicritical recipe of ref. [26] for supersymmetric JT gravity. Remarkably, this case is indeed non-perturbatively quite distinct from the $b<1$ models, in that it is intrinsically well-defined as matrix model, as can be seen in the original analysis of ref. [16], or from the properties of the string equation [19].

## B. Supersymmetric VMS: Background remarks and a 3D gravity connection

A guess [16] for a natural supersymmetric extension of the above construction is that the leading spectral density of a matrix model formulation of it would be a supersymmetric generalization of (1), such as, for an $\mathcal{N}=1$ CFT [27–31]:

$$\rho_0^{(b)}(E) = e^{S_0} 2\sqrt{2} \frac{\cosh(2\pi b\sqrt{E}) \cosh(2\pi b^{-1}\sqrt{E})}{\sqrt{E}} . \qquad (7)$$

describing the density of (NS-R) states with weight $h_p = \frac{(c-\frac{3}{2})}{24} + \frac{P^2}{2} + \frac{\delta}{16}$, with $\delta = 1$ (or 0) in the Ramond (or Neveu-Schwarz) sector. Again, $E = P^2$, $0 \leq b \leq 1$, and now $c = \frac{3}{2} + 3Q^2$, with $Q = b + b^{-1}$ in the spacelike (super)-Liouville theory. The timelike (super)-Liouville sector has $\hat{c} = \frac{3}{2} - 3\widehat{Q}^2$, $\widehat{Q} = b^{-1} - b$ and $\hat{h}_{\widehat{P}}$, such that $h_P + \hat{h}_{\widehat{P}} = 1$ and $c + \hat{c} = 15$.

Indeed, using this as a starting point, supersymmetric Virasoro minimal string theories were formulated as matrix models in ref. [20], and since an efficient fully non-perturbative approach was used (the technology of string equations, *etc.*), many features of the models were able to be extracted quite readily. In particular, for *all* values $0 \leq b \leq 1$ the models are non-perturbatively well-defined. A key step in ref. [20] was the identification of the precise combination of the relevant multicritical matrix models that makes up the supersymmetric model. This time:

$$t_k = 2\sqrt{2}\pi \frac{\pi^{2k}}{(k!)^2} \left( Q^{2k} + \widehat{Q}^{2k} \right) , \quad \mu = t_0 = 4\sqrt{2}\pi . \quad (8)$$

(The parameter $\mu$ will be explained in Sections IV and V). There is a striking comparison to be made with the bosonic VMS case (6), where it can be seen that there's a simple relative sign change between the $Q$ and $\widehat{Q}$ sectors. While this was observed in ref. [20], and its consequences explored (it completely changes the non-perturbative behaviour for example), it was not clear how to interpret it.

So for the first of several exciting new results in this paper, this sign change can be shown to have an understanding in a natural larger context: There is a 3D chiral *supersymmetric* gravity interpretation lurking, generalizing what was seen for the ordinary VMS! It works as follows. First note that the leading loop expectation value $Z_0$ for length $\beta$ that results from the Laplace transform of the density (7) is:

$$\begin{aligned} \hbar Z_0 &= \sqrt{\frac{2\pi}{\beta}} \left( e^{\frac{\pi^2 Q^2}{\beta}} + e^{\frac{\pi^2 \widehat{Q}^2}{\beta}} \right) \\ &= \sqrt{\frac{2\pi}{\beta}} e^{\frac{\pi^2 Q^2}{\beta}} \left( 1 + e^{-\frac{4\pi^2}{\beta}} \right) , \quad (9) \end{aligned}$$

which again seems only a slight modification of the VMS case by a swap of sign. How can this be understood?

Now, the natural $\text{SCFT}_2$ partition function would be expected to be built from NS and R sectors, and also with the inclusion of $(-1)^F$ in the trace. These are modular transforms of each other, so one representative copy will suffice, and (reading across from Maloney and Witten's 3D gravity work [24] [4]) what naturally presents itself is the $\text{Tr}_{\text{NS}}(-1)^F$ sector. So the natural vacuum analogue of (3) is:

$$Z(\tau) = \widetilde{q}^{-\frac{c}{24}} \prod_{n=2}^{\infty} \frac{1 - \widetilde{q}^{n-\frac{1}{2}}}{1 - \widetilde{q}^n} . \quad (10)$$

Again, things can be written in terms of $\eta(\widetilde{q})$, with the additional aid of:

$$\prod_{n=2}^{\infty} (1 - \widetilde{q}^{n+\frac{1}{2}}) = \widetilde{q}^{\frac{1}{48}} \frac{\eta(\tau/2)}{(1 - \widetilde{q}^{\frac{1}{2}})\eta(\tau)} , \quad (11)$$

and hence one can write:

$$\begin{aligned} Z(\tau) &= \widetilde{q}^{-\left(\frac{c}{24} - \frac{3}{48}\right)} \left( 1 + \widetilde{q}^{\frac{1}{2}} \right) \frac{\eta(\tau/2)}{\eta^2(\tau)} \\ &= \widetilde{q}^{-\frac{Q^2}{8}} \left( 1 + \widetilde{q}^{\frac{1}{2}} \right) \frac{\eta(\tau/2)}{\eta^2(\tau)} . \quad (12) \end{aligned}$$

Once again, doing an $S$-transform (using that $S : \eta(\tau/2) \to (2\tau_2)^{\frac{1}{2}} \eta(2\tau)$) and stripping off the $\eta-$functions leaves precisely the matrix model loop amplitude (9), if this time the identification $e^{-\frac{4\pi^2}{\beta}} \equiv q^{\frac{1}{2}}$ is made!

Turning to the trumpet partition function $Z_{\text{tr}}(\beta, P)$, the same story goes through as before. Exactly the same form (4) as before for the trumpet partition function appears for the SVMS (this will be confirmed in Section V [5]). Writing the $\text{Tr}_{\text{NS}}(-1)^F$ analogue of the primary character (4) yields, after using the supersymmetric formula for $h_P$, getting the same shift in overall powers of $\widetilde{q}$ as in the first line of (12), and $S$-transforming, the previous result $Z_{\text{tr}} = \sqrt{\frac{2\pi}{\beta}} q^{P^2}$ if once again $e^{-\frac{4\pi^2}{\beta}} \equiv q^{\frac{1}{2}}$ and the $\eta$-functions are removed.

Evidently, it is natural to conjecture that there are indeed 3D *supersymmetric* chiral gravity models that compute the same SVMS correlators as the matrix models with leading spectral density 7. There are at least two variants, 0A and 0B, and this paper will fully describe, refine, and compare the constructions of the matrix models to which they are (presumably) dual. The extension beyond just the leading disc sector is extremely natural: The solid torus becomes $\Sigma_{g,n} \times S^1$ on the 3D side, and the matrix models just do what comes naturally when it comes to computing to higher genus. Quite marvellously, a non-perturbatively well-defined matrix model dual (which is the case here) also endows a clean non-perturbative completion to the supergravity, neatly contained in the formalism in this paper. Later results about how the 0A and 0B matrix formulations are organized and related to each other in terms of string equation solutions (*etc.,*) could most likely be useful in understanding how to construct 0A and 0B 3D chiral supergravity models. This is all clearly worth investigating further!

### C. This paper: Purpose and main results

The SVMS models formulated non-perturbartively in ref. [20] were of 0A type, and in modern language,

---

[4] Ref. [32] was also found to be very useful.

[5] The SVSM trumpet appeared implicitly in the quantum volume calculations of ref. [20], but in unfortunate conventions.

fall into the $(2\Gamma+1, 2)$ Altland-Zirnbauer classification of ensembles, for a parameter $\Gamma$ that can be integer or half-integer. They reduce (as $b\to0$) to the analogous 0A-type supersymmetric JT models defined in ref. [13], which were defined and studied non-perturbatively in refs. [33, 34].

Crucially, it was observed that there is a pair of models (the cases $\Gamma=\pm\frac{1}{2}$) for which all orders in perturbation theory beyond leading order vanish identically, a non-trivial prediction for any continuum approach to the SVMS, whether it be super Liouville CFT, intersection theory, 3D chiral supergravity, or something else.

This paper will explain why the result (8) for the $t_k$, as well as the special $\Gamma=\pm\frac{1}{2}$ models, are *all that are needed* to define a natural 0B variant of the SVMS. This might seem counter to expectations, since the 0B matrix model should be expected to be a merged double-cut Hermitian matrix model (*i.e.*, a Dyson $\beta=2$ class of ensemble) – quite different from an AZ type ensemble! Nevertheless, combining the string equation solutions defining the special $\Gamma=\pm\frac{1}{2}$ models in one manner gives 0A matrix model solutions, but a different combination yields solutions of the appropriate string equations for 0B.

The 0B model also has all higher orders of perturbation theory vanish for loop correlators. Moreover, the non-perturbative corrections are severely suppressed, in comparison to the leading perturbative result, even when the string coupling is large. Of course, for $b\to0$ it reduces to the 0B JT supergravity that was defined in ref. [13] and formulated non-perturbatively in ref. [35].

Most of this could have been said explicitly in ref. [36], since all the key elements had already been computed, coupled with the obvious $b \to 0$ limit to the known 0A and 0B matrix model constructions of supersymmetric JT gravity [26, 35]. Nevertheless, there are several features and new results (such as those mentioned above) that seem worthwhile spelling out explicitly. Moreover, it may be of use to unpack the workings of some key elements of the double-scaled formalism that appears to have not been done in the literature before. This is also done in the hope that it might also serve as a useful guide to those working on continuum approaches to supersymmetric Virasoro strings and variants of them.

Here is a detailed summary of this paper's key results:

- The first new results are **Section I B**'s interpretation (above) of the matrix model's leading loop amplitude (9), and trumpet partition function (4) as $\text{CFT}_2$ S-transformed vacuum and primary characters (presumably arising from a dual 3D chiral supergravity). That such matrix models can be naturally defined, and moreover that they naturally define a series of higher genus results for the $\Sigma_{g,n}\times S^1$ solid torus (with both 0A and 0B variants) opens up a fascinating avenue of research.

- **Section II** reviews the core of the matrix model 0A and 0B constructions in terms of the different sets of string equations that arise after double scaling.

The 0A KdV-associated string equation defines a function $u(x)$ while the 0B mKdV-associated [6] equations define a function $r(x)$. While they seem very different from each other in some respects, it was observed long ago [40] that there is a relation between them through the Miura map $u=r^2+\hbar r'$. New aspects of this map, when focussed on the special $\Gamma=\pm\frac{1}{2}$ solutions, emerge in this paper in **Section III** and **Section IV**, and are explored explicitly in terms of worldsheet expansions provided by each matrix model. When there is a choice to be made, the difference between 0A and 0B constructions is simply this: The matrix models derive the string worldsheet expansion from a function that is, in essence, an *average* of two functions picked from the pair $\{u_{+\frac{1}{2}}, u_{-\frac{1}{2}}\}$. If they are chosen to match each other (*i.e.*, correlating the $\pm\frac{1}{2}$ choices), then one or other of the special 0A models results. If intead the choice averages over the $\pm\frac{1}{2}$, the physics of 0B results. Such *string equation*-level choices for construction of theories evidently *fully mirrors* the choices (involving $(-1)^F$ grading) concerning worldsheet spin structures in continuum CFT approaches (and pin structures, because the special 0A models are non-orientable), and seems to not have been observed before.

- **Section IV** also reviews a crucial CFT property, first observed in ref. [39], of the combined torus partition functions of the $k$th multicritical models on the 0A and 0B sides: $Z_{\text{even}}\equiv\frac{1}{2}(Z_A+Z_B)=-\frac{1}{16}\ln|x|$. It then goes further to verify, using Section IV's relation between 0A and 0B string equation solutions, that the CFT relation holds for *any* admixture of multicritical models (defined by the $t_k$). This is non-trivial since the string equations are highly non-linear.

- **Section V** begins by reviewing the connection (through underlying integrable KdV flows of the function $u(x)$) between point-like and macroscopic loop observables in 0A. The fact [38] that 0B loop operators are constructed as a sum over two sectors is then neatly derived from the fact that two separate (mKdV) flow structures arise from the distinct pair $u_{\pm\frac{1}{2}}(x)$ from which the 0B system is built. Some examples of the quantum volumes are computed in the matrix model, and some general results that follow robustly from the underlying formalism are presented. That the $u_{\pm\frac{1}{2}}(x)$ vanish to all orders in $\hbar$ in the perturbative regime immediately implies the same for the 0B system built by

---

[6] They are part of a larger (Zakharov-Shabat-associated) system of equations [37–39], but that structure only appears when certain background R-R sources are turned on, which won't be done here.

summing them. In particular, all "quantum volumes" (in the sense of ref. [16]) are identically zero, a non-trivial prediction for the various dual continuum approaches.

- **Section VI** reviews the process of matching the 0A leading string equation to the leading spectral density (7) yielding formula (8) for the $t_k$, and shows why the leading equation for 0B yields precisely the same $t_k$ (this is required by the observations of Section IV, incidentally). Finally, mirroring the case done for 0A SVMS in ref. [20], the non-perturbative spectral density is computed for 0B SVMS. Strikingly, the non-perturbative undulations naively expected are highly washed out. This fits with intuition for a model that has no classical spectral edge [7] In terms of the key aspects of the construction, it all follows from the fact that the *separate* non-perturbative undulations of the $\Gamma = \pm\frac{1}{2}$ pair are almost completely out of phase with each other and so (nearly) cancel when summed to make 0B.

## II.  TWO CLASSES OF MULTICRITICALITY

The starting point for formulating the desired matrix models of the string theories of interest is to look for examples of multicriticality that concern random matrices of the appropriate class. The underlying holographic dual should have a positive energy spectrum for its Hamiltonian $H$ (as naturally occurs in type 0A and 0B models). Here, $H$ can be written in terms of a supercharge as $H=Q^2$. (There is an unfortunate notational clash with $Q$, the Liouville parameter, but hopefully context will make meanings clear henceforth.)

There are two broad classes of model [13]: There is either a $(-1)^F$ symmetry (type A), or there is not (type B). In the first case, in a representation of $(-1)^F$ as block diagonal, $Q$ can be written as:

$$Q = \begin{pmatrix} 0 & M \\ M^\dagger & 0 \end{pmatrix} \ , \tag{13}$$

where $M$ is a complex matrix. In the second case, where there is no additional constraint on $Q$, it is simply Her-

---

[7] This was anticipated in ref. [13] for 0B JT supergravity and confirmed in ref. [35]. The point is that significant undulations stem from the presence of endpoints/zeros of the leading spectral density, but the 0B system, being a merged two-cut system, has no such endpoints. At a more detailed level, the undulations are a feature of the underlying probability peaks for individual energy levels across the ensemble of matrices [41]. Peaks can spread and become more distinct from their neighbours the closer they are to an endpoint, since there is less repulsion from other peaks on one side compared to the other. Peaks in the bulk of an overall spectrum, by contrast, are squeezed from both left and right and so are narrower and less distinct form each other. The resulting undulations far from endpoints are therefore less marked.

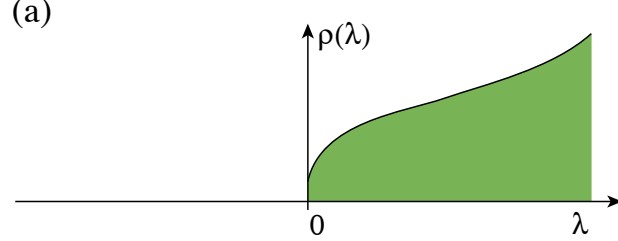

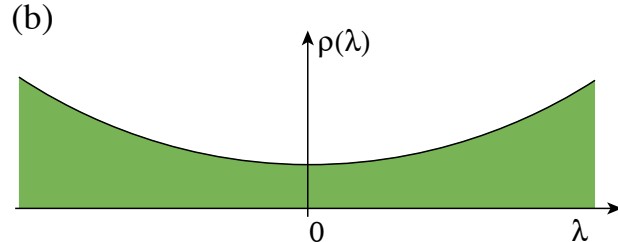

Figure 2. The two classes of behaviour upon which the multi-critical models are based: (a) Hard-edge behaviour where the eigenvalue distribution collides with a wall. (b) Double-cut behaviour where two symmetric components of the eigenvalue distribution have collided and merged.

mitian. The matrix model multicriticality is then characterized nicely by what is happening at the (putative) endpoints of their large $N$ eigenvalue distribution. These are described in the next two Subsections, appropriately named **A** and **B**.

### A.  "Hard edge" behaviour.

The $M^\dagger M$ (or $MM^\dagger$) eigenvalues $\lambda$ bump into a wall (at $\lambda=0$) corresponding to the fact that the parent ensemble is of positive matrices. See Figure 2(a). The basic prototype model is of "hard-edge" Wishart type [42], and can be generalized by having a potential tuned such that there are $k$ additional zeros of $\rho_0(\lambda)$ at $\lambda=0$. Such multicritical models were discovered and explored in refs. [40, 43, 44], and are now classified as type $(2\Gamma+1, 2)$ in the $(\boldsymbol{\alpha}, \boldsymbol{\beta})$ Altland-Zirnbauer classification [45] of ensembles based on the form of the engenvalue measure. Here $\Gamma$ is integer or half-integer and is a natural parameter to be discussed later.

Matrix models are efficiently solved by formulating them in terms of families of orthogonal polynomials, typically indexed by an integer, $n$. The first $N$ polynomials can be used as a basis for building matrix model observables. The potential of the matrix model determines the family, through recursion relations whose recursion coefficients satisfy difference equations. At large $N$, in the double scaling limit, the recursion coefficients become functions of a continuous parameter $x$ that began life as the orthogonal polynomial index $n$. The difference equations in turn become non-linear ordinary differential

equations, the "string equations" of the model. In the case of the A-type behaviour described above, there is one function denoted $u(x)$ and the equation is [40, 43]:

$$u\mathcal{R}^2 - \frac{\hbar^2}{2}\mathcal{R}\mathcal{R}'' + \frac{\hbar^2}{4}(\mathcal{R}')^2 = \hbar^2\Gamma^2 , \qquad (14)$$

with $\mathcal{R} \equiv \sum_{k=1}^{\infty} t_k R_k[u]+x$ where the $R_k[u]$ are polynomials (see below) in $u(x)$ and its $x$-derivatives, normalized here so that the purely polynomial part is unity, i.e., $R_k = u^k + \cdots \# \hbar^{2k-2}u^{(2k-2)}$, where $u^{(m)}$ means the $m$th derivative. The intermediate terms involve mixed orders of derivatives, and every derivative comes with an $\hbar$. The first couple are $R_1[u]=u$ and $R_2[u]=u^2-\frac{\hbar^2}{3}u''$. General $R_k$ can be determined by a recursion relation, but it will not be needed here.

The $t_k$ are coefficients that depend on the model. They control how much of each multicritical model contributes to describing the gravity model in question. Ref. [20] established that for the type 0A supersymmetric Virasoro minimal string (SVMS), the relevant combination is given by equation (8). This will be rederived in Section V and then shown to also be just what is needed for supersymmetric type 0B SVMS.

## B. "Double cut" behaviour

Models of the Dyson-type classification [42, 46] (labeled by $\boldsymbol{\beta}$, which is 2 here since $Q$ is Hermitian), generically have "soft" edges for the spectral density endpoints. However, the potential can tuned so that there are two separate parts of the density/distribution of eigenvalues $\lambda$ of the Hermitian $Q$. A distinct type of criticality arises when their separate "soft" edges collide with each other. A simple example is the symmetric case where the merging components are mirror reflections of each other (but more general asymmetric arrangements are possible too [47], see below.) In this case, the basic prototype is the local physics of the Gross-Witten-Wadia [48, 49] phase transition. The original context is a unitary matrix model, where the $\lambda$ live on a circle. The phase transition occurs when going from filling only part of the circle to completely filling it after the ends collide. Since the universal physics is just in the neighbourhood of the ends, this shares the same physics as the symmetric two-cut Hermitian case. See Figure 2(b) for a sketch of a post-collision merger.

Multicritical generalizations where the potential is tuned to add extra zeros at the collision were first found by Periwal and Shevitz [50, 51] (with an extension by [52] to add extra "quark" flavours). In these models, the relevant string equation can be written in terms of a function $r(x)$ and it is:

$$\sum_{k=1}^{\infty} t_{2k}K_{2k}[r] + rx + \hbar C = 0 , \qquad (15)$$

where the $K_{2k}[r]$ are differential polynomials in the $r(x)$ and its derivatives, with normalization $K_{2k}=r^{2k+1}+\cdots+\#\hbar^{2k}r^{(2k)}$ with mixed derivative intermediate terms as before. The $t_{2k}$ determine the admixture of multicritical models that determine the gravity/string theory. They will be determined shortly. The set of equations (15), is the Painlevé II hierarchy of ordinary differential equations (ODEs), getting its name from the case $t_{2k}=\delta_{k,1}$. There, since $K_2=r^3-\hbar^2 r''/2$, the equation is the classic Painlevé II:

$$-\frac{\hbar^2}{2}r'' + r^3 + rx + \hbar C = 0 . \qquad (16)$$

The case $C=0$ will be of particular interest. In that case, the equation for the leading solution, $r_0(x)$ is $r_0(r_0^2+x)=0$. The solution $r_0=0$ is appropriate to expanding in the $x\to+\infty$ direction, and developing perturbation theory around that starting point results in all corrections being zero. There are only non-perturbative (in $1/|x|$ or $\hbar$) corrections in this region (see later for more on this). The solution $r_0=\sqrt{-x}$ is the seed for a perturbative expansion in the $x\to-\infty$ direction and the first few terms are:

$$r(x) = \sqrt{-x} \qquad\qquad (17)$$
$$+ \frac{h^2}{16(-x)^{\frac{3}{2}}x} - \frac{73h^4}{512(-x)^{\frac{11}{2}}} - \frac{10657h^6}{8192(-x)^{\frac{17}{2}}} + \cdots$$
$$r^2(x) = -x - \frac{h^2}{8x^2} + \frac{9h^4}{32x^5} - \frac{1323h^6}{512x^8} + \cdots , \; x \to -\infty ,$$

where it will later (Sections IV and V) be useful to consider how things assemble into $r(x)^2$, since it naturally defines the free energy of the 0B theory. That this expansion is really that of a string theory worldsheet (or 2D gravity theory) was recognized early on in ref. [40] because of a relation (see Section III) to the solutions to (14). It was later fully recognized as part of the story of type 0A and 0B in ref. [39]. More about worldsheet expansions will be explored in Section IV.

For all $k$, the case $C=0$ is what arises from the simple symmetric two-cut $\beta=2$ Dyson ensemble (i.e., Hermitian matrix models with certain even potentials), and that will be the focus here, although several of the observations to be made later have a natural generalization to include $C$, and interpretations should be sought.[8]

There is a different way of introducing an important constant into the equations, by considering more general double cut Hermitian matrix models. The string equations are enlarged to the Zakharov-Shabat hierarchy of equations [37, 38, 47, 53, 54] involving another

———

[8] In the unitary matrix model context the constant $C$ has a natural interpretation as counting the number of quark flavours in the fundamental representation, but it can just as well be thought of a associated to a certain kind of external field in the hermitian matrix model [52].

function $\alpha(x)$:

$$\sum_{m=1}^{\infty} t_m K_m[r,\alpha] + xK_0 = 0 \ ,$$

$$\sum_{m=1}^{\infty} t_m H_m[r,\alpha] + xH_0 + \hbar\Upsilon = 0 \ , \qquad (18)$$

using the presentation of ref. [39] (with some notation changes[9]) where $\Upsilon$ is a constant (to be discussed below), here $m$ in general is odd or even. Now, $K_m[r,\alpha]$ and $H_m[r,\alpha]$ are differential polynomials in both $r(x)$ and $\alpha(x)$, normalized in a manner consistent with the even case given below (15). They satisfy a recursion relation giving an infinite tower of them with $K_0=r(x)$ and $H_0=0$. For the case of even $m=2k$ and $\Upsilon=0$, the $H_{2k}=0$ and the function $\alpha(x)$ becomes a constant that can be shifted away, and the system returns to the form given in (15). Non-zero $\Upsilon$ captures new physics. For example the $k=1$ model has $K_2=r^3 - \hbar^2 r''/2 - 2r\alpha^2$ and $H_2=2r^2\alpha$. Eliminating $\alpha$, one can write [55]:

$$-\frac{\hbar^2}{2}r'' + rx + r^3 - \frac{\hbar^2\Upsilon^2}{2r^3} = 0 \ , \qquad (19)$$

so $\Upsilon$ can be thought of [37, 38, 47] as controlling a deformation of the Painlevé II form, this time by an inverse $r^3$ term. This will be returned to later.

Henceforth, the case of even $m=2k$ will be the focus, corresponding to cases where a well-defined bounded potential can be written. The cases of odd $m$ can be thought of as perturbations of the even $m$ cases [38, 39, 53, 54], but will not be a focus for most of the rest of this work.

### III. A POWERFUL MAP–THE VANISHING

While the two families of string equation (14), and (15) seem quite different, with different origins, there is a relation between them, found long ago in ref. [40]. With a solution $u_\Gamma(x)$ to equation (14), if one writes $u=r^2+\hbar r'$, then $r(x)$ satisfies (15), with $C=\frac{1}{2}\pm\Gamma$. Call these solutions $r_C$ and $r_{1-C}$. The fact that there are two choices is important. Equation (15) is symmetric under exchanging $r\to -r$ and $C\to -C$, so solutions $r_{-C}=-r_C$. This means that if one began again with (14) with some different value, $\Gamma'$, and instead formed $u_{\Gamma'}=r^2-\hbar r'$, one would arrive at two copies of (15) for some $-C'$ and $C'-1$ with solutions $v_{-C'}$ and $v_{C'-1}$, where $C'=\frac{1}{2}\pm\Gamma'$. Now consider the case $\Gamma'=\Gamma\pm1$. That would result in either the solution pair $(\Gamma+\frac{3}{2},\Gamma+\frac{1}{2})$ or $(\Gamma-\frac{3}{2},\Gamma-\frac{1}{2})$. So the case of interest, $C\equiv\frac{1}{2}\pm\Gamma=0$ could have come from either starting with a $\Gamma=\frac{1}{2}$ solution and constructing $r$ from $u_{\frac{1}{2}}=r^2+\hbar r'$ or starting with a $\Gamma=-\frac{1}{2}$ solution and instead doing $u_{-\frac{1}{2}}=r^2-\hbar r'$.

---

[9] $K_m,\alpha(x)$, and $\Upsilon$ here are (respectively) $R_m,\beta(x)$, and $q$ there.

These $u_{\pm\frac{1}{2}}$ are distinct solutions of the type 0A equation (14) with different physical properties (see for example figure 7). On the other hand it is noticeable that the string equation (15) does not distinguish between the two. Indeed it cannot, and so rather than pick one or the other, the 0B system is built by having equal contributions from both, as will be explained in Section V. Indeed, this situation sounds a lot like the difference between 0A and 0B in continuum approaches: 0A cares about the grading in terms of $(-1)^F$, and 0B does not. All the evidence suggests that this is not accidental.

There is a very special property of all solutions of the two families of string equations for $C=0$ i.e., $\Gamma=\pm\frac{1}{2}$. It can be seen as follows. Perturbation theory beyond leading order is developed by expanding $u(x)$ around either of the $x\to+\infty$ or $x\to-\infty$ regimes. The leading solution for $x>0$ regime is zero, followed by a universal term at order $\hbar^2$: $u(x)=0+\hbar^2\left(\Gamma^2-\frac{1}{4}\right)/x^2+\cdots$, but crucially, upon recursively solving to higher orders, all further terms in the expansion come with the $\left(\Gamma^2-\frac{1}{4}\right)$ prefactor:

$$u(x) = \left(\Gamma^2 - \frac{1}{4}\right)\sum_{g=1}^{\infty}\hbar^{2g-2}\frac{U_g}{x^{2g}} + \text{non-perturb.}, \quad (20)$$

where $U_1=1$, and for $g>1$ the constants $U_g$ depend on the details of the particular model. So in either case of $\Gamma=\pm\frac{1}{2}$, the solution for $u(x)$ vanishes to all orders in perturbation theory. This translates into the $r(x)$ that it maps into under $u=r^2\pm\hbar r'$, which solves (15) with $C=0$, i.e., the solution for $r(x)$ also has this all-orders vanishing property! This is a key feature of the type 0B solutions of interest, and of course of the $(\Gamma=\pm\frac{1}{2})$ type 0A theories built on $u_{\pm\frac{1}{2}}$ already defined and studied in ref. [20].

### IV. STRING WORLDSHEET PICTURE

Much can be learned directly about the string theories from expanding the string equation solutions $u(x)$ and $r^2(x)$, and remembering that (after integrating twice with respect to $x$) they give the free energy $F(\mu)$ of the string theory. For 0A the precise relation is:

$$2\hbar^2\frac{\partial^2 F}{\partial\mu^2}\bigg|_{\mu=x} = u(x) \ . \qquad \text{(0A)} \qquad (21)$$

Given what was learned in the previous Section, it is natural to study the nature of the $u_{\pm\frac{1}{2}}$ solutions to uncover precisely why and how they yield a closed string theory. This will nicely demonstrate using this approach. The key relation that will emerge is that the 0B function will be costructed as $r^2=\frac{1}{2}[u_{\frac{1}{2}}+u_{-\frac{1}{2}}]$, and this fixes the relation between the free energy and the function $r(x)$ for 0B:

$$\hbar^2\frac{\partial^2 F}{\partial\mu^2}\bigg|_{\mu=x} = r^2(x) \ , \qquad \text{(0B)} \qquad (22)$$

and it is interesting to see the consequences of these relations flow through to various aspects of the theories.

## A.   Warmup: Pure Supergravity

For definiteness, consider the $k{=}1$ (pure supergravity) example of the 0A theory. For a given $\Gamma$, one can expand the first few terms in the $x \to \pm\infty$ regimes to get:

$$u_\Gamma(x) = \left(\Gamma^2 - \frac{1}{4}\right) \frac{\hbar^2}{x^2} \left\{ 1 - \frac{2\hbar^2 \left(\Gamma^2 - \frac{9}{4}\right)}{x^3} \left( 1 - \frac{7\hbar^2 \left(\Gamma^2 - \frac{21}{4}\right)}{8x^3} \right) + \cdots \right\} \qquad (x \to +\infty), \quad (23)$$

$$u_\Gamma(x) = -x \pm \frac{\hbar\Gamma}{\sqrt{-x}} - \frac{\hbar^2\Gamma^2}{2x^2} \pm \frac{5\hbar^3\Gamma \left(\Gamma^2 + \frac{1}{4}\right)}{8(-x)^{\frac{7}{2}}} + \frac{\hbar^4\Gamma^2 \left(\Gamma^2 + \frac{7}{8}\right)}{x^5}$$

$$\pm \frac{11\hbar^5\Gamma \left(336\Gamma^4 + 664\Gamma^2 + 105\right)}{2048(-x)^{\frac{13}{2}}} - \frac{7\hbar^6\Gamma^2 \left(256\Gamma^4 + 932\Gamma^2 + 507\right)}{512x^8} + \cdots \qquad (x \to -\infty). \quad (24)$$

For the first expansion, there are even powers of the topological coupling $\hbar$ only, consistent with the fact that this is a closed string theory with no boundaries on the worldsheets. Integer $\Gamma$ has the interpretation as units of R-R flux. See Figure 3. On the other hand, the $x{<}0$ regime

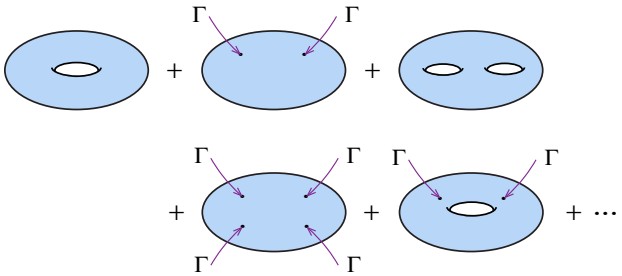

Figure 3. Some of the worldsheet closed string diagrams with flux insertions, associated to the $x{>}0$ regime in (23).

has both even and odd powers of $\hbar$, and in a topological interpretation, these are Riemann surfaces again but now a factor of $\Gamma$ appears with each boundary. See Figure 4. Integer $\Gamma$ then naturally has an association with

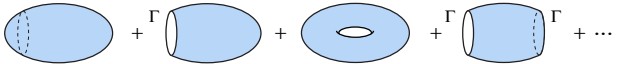

Figure 4. Some of the worldsheet diagrams with boundaries associated to the $x{<}0$ regime in (24).

D-branes, carrying a total of $\Gamma$ units of R-R charge. Notice that in this regime there are two choices, generated by the $\pm\Gamma$ choice that entered in taking a square root in the order $\hbar$ term. This correlates with a sign choice for the charge. For brevity in the below (and already in Section III above), a notation will be used in a subscript on function $u(x)$ that denotes $\pm\Gamma$ choices.

The cases of interest in this paper will be $\Gamma{=}0$ and $\Gamma{=}\pm\frac{1}{2}$. These correspond to $(1,2)$, $(0,2)$ and $(2,2)$ AZ ensembles, because $\boldsymbol{\alpha}{=}2\Gamma{+}1$ and $\boldsymbol{\beta}{=}2$. The latter two cases have half-integer $\Gamma$, and are in fact to be better understood [13] as unorientable theories, *i.e.*, rather than counting boundaries, $\Gamma$ counts the insertions of crosscaps (*i.e.*, gluing in an $\mathbb{RP}^2$–See Figure 5). Such surfaces can



Figure 5. Some unorientable closed string diagrams made by including crosscaps, associated to the $x{>}0$ expansion (23) when $\Gamma{=}\pm\frac{1}{2}$.

come with either sign. A particular 0A theory uses either one choice of sign or the other. This is equivalent to taking, as the solution for $u(x)$, the average of two copies of $u_{\pm\frac{1}{2}}(x)$, *i.e.*, where the sign choices are chosen to match. Spelling it out (perhaps unnecessarily):

$$u^{0A}_{\pm\frac{1}{2}} = \frac{1}{2}[u_{\pm\frac{1}{2}} + u_{\pm\frac{1}{2}}]. \qquad (25)$$

It is natural then to consider the alternative situation where the sum is taken over the choices without regard to the sign:

$$u^{0B} = \frac{1}{2}[u_{\pm\frac{1}{2}} + u_{\mp\frac{1}{2}}] = r^2, \qquad (26)$$

(where the label on $u$ heralds the conclusion to come). In the perturbative $(x{>}0)$ regime, every order in perturbation theory still vanishes in the separate $u(x)$, but in the $x{<}0$ regime something interesting happens instead. The terms with odd powers of $\hbar$ vanish, *i.e.* all the crosscap sectors (or for general $\Gamma$, open string sectors) vanish in

taking the combination, leaving:

$$u_{+\frac{1}{2}} + u_{-\frac{1}{2}} = -2x - \frac{\hbar^2}{4x^2} + \frac{9\hbar^4}{16x^5} - \frac{1323\hbar^6}{256x^8} + \cdots \,, \quad (27)$$

a purely closed and orientable string worldsheet expansion (Figure 6).



Figure 6. Some of the purely closed and orientable string diagrams for expansion (27).

Strikingly this is what is obtained by expanding in the $x<0$ direction the solution for $2r(x)^2$ of Painlevé II, see equation (17). This is already identified with the $k=1$ type 0B theory, and a natural conclusion presents itself.

The two kinds of choice, (25) and (26), for what to do with the expansions (23) should be viewed as the direct double-scaled matrix model manifestation of the continuum constructions of 0A $vs$ 0B. The issue is whether one sums over supermanifolds taking into account weighting by $(-1)^\zeta$ or not, where $\zeta$ is correlated with fermion number in the dual (boundary) theory. The sign choices in front of crosscaps ($\mathbb{RP}^2$) in turn correlates with $\zeta$, which tracks the pin$^-$ structure of the manifolds–See ref. [13].

Moreover, after sending $x \rightarrow -2^{-\frac{1}{3}}x$ the expansion (27) becomes:

$$2^{\frac{1}{3}}r^2 - x = \left( -\frac{\hbar^2}{4x^2} - \frac{9\hbar^4}{8x^5} - \frac{1323\hbar^6}{64x^8} + \cdots \right) \,, \quad (28)$$

but this can be recognized as the $+x$ perturbative expansion (23) of the 0A theory after setting $\Gamma = 0$. So the $x < 0$ perturbative physics of 0B for $k = 1$ can be described in terms of a dual closed string theory which is 0A theory in the $x > 0$ direction. Indeed, it has been long known [56] that the $k=1$ string equation for $u(x)$ with parameter $\Gamma = 0$ maps, upon identifying $u = 2\tilde{r}^2 - x$, ($i.e.,$ a different map from the one in Section III!) to Painlevé II with $C = 0$. Using that map with $\Gamma \neq 0$ yields [55]:

$$-\hbar^2 \frac{\tilde{r}''}{2} - \frac{\tilde{r}x}{2} + \tilde{r}^3 - \frac{\hbar^2\Gamma^2}{8\tilde{r}^3} = 0 \,, \quad (29)$$

and writing $\tilde{r} = 2^{-\frac{1}{3}}r$ and sending $x \rightarrow -2^{\frac{1}{3}}x$ gives the deformed Painlevé II equation found earlier (19) for 0B with $\Upsilon=\Gamma$. The $\Gamma=0$ case then reproduces the observation concerning (27) about $u_{+\frac{1}{2}}+u_{-\frac{1}{2}}$.

For general $\Gamma$ and $\Upsilon$, with $k=1$, the 0A and 0B theories are dual [55] to each other as closed string theories under the exchange $-x \leftrightarrow x$, and $\Gamma$ and $\Upsilon$ exchange roles in counting R-R fluxes and D-branes.

For $k>1$ the magical fact that the pairs of closed string sectors for the 0A and 0B theories map to each other under exchange of $x$ with $-x$ goes away, but it is generally true that, starting with the expansion of $u(x)$ in the $-x$ regime, one will always get closed string sum over worldsheets if one adds together expansions from any $\pm\Gamma$ pair, although $\pm\frac{1}{2}$ is special, for the reasons outlined above.

## B. The torus partition function, and the $k$th model

The different expansions in the $\pm$ regimes for each set of equations are still tightly correlated, however, as is clear from an important observation [55] concerning the one-loop free energy computed in conformal field theory:

$$Z_{\text{even}} \equiv \frac{1}{2}\left(Z_A + Z_B\right) = -\frac{1}{16}\ln|x| \,, \quad (30)$$

and for the $-x$ expansion of the 0A equations the prefactor is $-\frac{k-1}{24k}$, while for 0B the $-x$ result is $-\frac{2k+1}{24k}$, which confirms the CFT result (30). Meanwhile the $+x$ expansions give $-\frac{1}{8}$ and 0 respectively, also confirming (30). See table I for a summary.

| | $x < 0$ | $x > 0$ |
|---|---|---|
| 0A | $-\frac{k-1}{24k}\ln(-x)$ | $-\frac{1}{8}\ln(x)$ |
| 0B | $-\frac{2k+1}{24k}\ln(-x)$ | $0$ |

Table I. Summary of the torus results for 0A and 0B in the two perturbative regimes for the general $k$ case. Adding vertically confirms equation (30).

Note that these formulae gives for $k = 1$ the pair $(-\frac{1}{8}, 0)$ that get exchanged. See Table II.

| | $x < 0$ | $x > 0$ |
|---|---|---|
| 0A | $0$ | $-\frac{1}{8}\ln(x)$ |
| 0B | $-\frac{1}{8}\ln(-x)$ | $0$ |

Table II. Summary of the torus results for 0A and 0B in the two perturbative regimes for pure supergravity, $k = 1$, which are dual pairs. Adding vertically confirms equation (30).

This tight correlation between expansions for each $k$ is extremely important here since it means that any combination of the models, given by a set of $t_k$s, used to build a 0A model $must$ $remain$ $in$ $that$ $combination$ to build a consistent corresponding 0B theory. So the set (8) already identified in ref. [20] for defining the 0A class supersymmetric VMS $must$ $also$ define the 0B class. This will also emerge naturally from the loop and spectral density construction later in Section VI.

## C. The torus partition function for 0A/0B supersymmetric VMS, and beyond

Since the string equations for 0A and 0B are non-linear, the results for the previous section for each individual $k$ do not neccessarily follow for models built by combining the $k$th models, which is essential for defining minimal string models, and the supersymmetric Virasoro minimal string models that are the focus here.

This section shows that it is possible to explicitly compute the result (30) for the full string equation with the

infinite set of $t_k$s turned on. The procedure will be to take each $(\pm x)$ perturbative direction in turn and derive the function $u_2(x)$ in the expansion $u(x) = \sum_{g=0} u_{2g}\hbar^{2g} + \cdots$ that controls the torus contribution.

The result for the $+x$ direction is straightforward since for 0A it has already been noted that the $-\frac{1}{4x^2}$ is universal (it yields the $-\frac{1}{8}$ after dividing by 2 (see equation (21)) and integrating, and for 0B the answer is 0 since all perturbative contributions vanish. More challenging is finding a closed form for the $-x$ regime. The relevant computation for 0A can be adapted from the random matrix model work on the ordinary VMS in ref. [19], where it was shown that:

$$u_2 = \frac{1}{12}\left[-2\frac{\ddot{G}^2}{(\dot{G})^4} + \frac{\dddot{G}}{(\dot{G})^3}\right] . \qquad (31)$$

Here, $G[u_0] \equiv \sum_k t_k u_0^k$ and an overdot denotes $\frac{d}{du_0}$. As a check, by putting $G = u_0^k$ to yield the $k$th model, a little algebra shows that (31) reduces to $-\frac{k-1}{12kx^2}$.

Note that in terms of $G$, the leading string equation for $x<0$ is written as $G[u_0]+x=0$. For later use, note that this means that $u_0$ derivatives and $x$ derivatives (denoted with a prime) can be interchanged using $e.g.$:

$$u_0' = -\dot{G}^{-1} , \quad u_0'' = -\dot{G}^{-3}\ddot{G} , \quad \cdots \qquad (32)$$

Before proceeding it will be useful to recall how (31) was derived. The key point is that in the $x<0$ regime, perturbation theory for the string equation (14) with $\Gamma=0$ is equivalent to perturbation theory for the simpler equation $\mathcal{R}=0$, $i.e.$:

$$\sum_{k=1}^{\infty} t_k R_k[u] + x = 0 . \qquad (33)$$

The $k$th polynomial $R_k[u]$ can be expanded in even powers of $\hbar$, and the leading two terms are [35]:

$$R_k = u^k - \frac{\hbar^2}{12}k(k-1)u^{k-3}\left[2uu'' + (k-2)(u')^2\right] + \cdots \quad (34)$$

Writing $u(x) = u_0(x) + \hbar^2 u_2(x) + \cdots$ gives:

$$\sum_k t_k u^k = \sum_k t_k[u_0^k + \hbar^2 k u_0^{k-1} u_2 + \cdots]$$
$$= G + \hbar^2 u_2 \dot{G} + \cdots , \qquad (35)$$

(which amounts to the first term in the Taylor expansion of $G[u]$). Placing this, with the $\hbar^2$ terms from (34), into equation (33) and using (32) yields the result (31).

In fact, because $\dot{G}=-1/u_0'$, the object in braces in (31) can be $x$-integrated twice to give $-\ln(-u_0')$, and so, after

dividing by 2 the 0A torus result is:

$$Z_A = \frac{1}{24}\ln(-u_0') . \qquad (36)$$

Turning to the case of 0B, it is most efficient to use the map discussed in Sections III and IV, first deriving the general form of the leading terms for the expansion of $u_\Gamma(x) = u_0(x) + \hbar u_1(x) + \hbar^2 u_2(x) + \cdots$ in the $x<0$ regime[10] now for non-zero $\Gamma$, and then constructing $\frac{1}{2}(u_{\frac{1}{2}} + u_{-\frac{1}{2}})$, which will give the expansion of $r^2$ needed for 0B. This approach will naturally give everything in terms of $G(u_0)$ and $u_0$ again, to make straightforward the process of combining with the 0A result. The derivation supplements what was done above for the $\Gamma=0$ case, and uses the fact that the $x<0$ perturbative expansion of the big string equation (14) is actually equivalent to $x<0$ expanding the following [57, 58]: $\mathcal{R}=2\hbar\Gamma\widehat{R}[u]$, $i.e.$:

$$\sum_{k=1}^{\infty} t_k R_k[u] + x = 2\hbar\Gamma\widehat{R}[u] . \qquad (37)$$

Here, the function $\widehat{R}[u]$ is the $\sigma=0$ limit of the object $\widehat{R}(\sigma,x) = \hbar\langle x|(\mathcal{H}-\sigma)^{-1}|x\rangle$, the diagonal resolvent of $\mathcal{H}$, which solves:

$$4[u(x) - \sigma]\widehat{R}^2 - 2\hbar^2\widehat{R}\widehat{R}'' + \hbar^2(\widehat{R}')^2 = 1 . \qquad (38)$$

Needed here is just the leading solution:[11] $\widehat{R}=\frac{1}{2}(u(x)-\sigma)^{-\frac{1}{2}}+\cdots$, and substituting the $\hbar$-expansion of $u(x)$ gives:

$$\widehat{R} = \frac{1}{2}\frac{1}{[u_0(x)-\sigma]^{\frac{1}{2}}} - \frac{1}{4}\frac{\hbar u_1(x)}{[u_0(x)-\sigma]^{\frac{3}{2}}} + \cdots \qquad (39)$$

Meanwhile, there are extra terms coming from $u_1$ in:

$$\sum_k t_k u^k = \sum_k t_k[u_0^k + k u_0^{k-1}(\hbar u_1 + \hbar^2 u_2)$$
$$+ \frac{k(k-1)}{2}\hbar^2 u_1^2 u_0^{k-2} + \cdots]$$
$$= G + \dot{G}(\hbar u_1 + \hbar^2 u_2) + \frac{\hbar^2}{2}u_1^2\ddot{G} + \cdots , (40)$$

(again, amounting to the Taylor expansion of $G[u]$ keeping up to order $\hbar^2$) and this yields, from order $\hbar$ terms in (37), $u_1 = \frac{\Gamma}{u_0^{1/2}\dot{G}}$. Meanwhile, at order $\hbar^2$, the $\Gamma = 0$ result (31) is supplemented by two additional terms, from the last term of (40) and the other from $2\hbar\Gamma$ times the last term in (39). The final result is:

_______

[10] The order $\hbar$ term $u_1(x)$ is there since $\Gamma$ is turned on.

[11] See refs. [19, 20, 59] for additional terms.

$$u_\Gamma(x) = u_0(x) + \hbar \frac{\Gamma}{u_0^{1/2}\dot{G}} + \hbar^2 \left\{ -\frac{1}{2}\Gamma^2 \left[ \frac{1}{u_0^2\dot{G}^2} + \frac{\ddot{G}}{u_0\dot{G}^3} \right] + \frac{1}{12} \left[ -2\frac{\ddot{G}^2}{\dot{G}^3} + \frac{\dddot{G}}{\dot{G}^4} \right] \right\} + \cdots , \tag{41}$$

and so forming $\frac{1}{2}(u_{\frac{1}{2}} + u_{-\frac{1}{2}})$ gives:

$$r^2(x) = r_0^2(x) + \hbar^2 \left\{ -\frac{1}{8} \left[ \frac{1}{u_0^2\dot{G}^2} + \frac{\ddot{G}}{u_0\dot{G}^3} \right] + \frac{1}{12} \left[ -2\frac{\ddot{G}^2}{\dot{G}^3} + \frac{\dddot{G}}{\dot{G}^4} \right] \right\} + \cdots , \tag{42}$$

where $r_0^2 = u_0$, and the 0B leading string equation is $G(r_0^2)+x=0$, which is $\Gamma$ independent. A quick check shows that for the $k$th model, *i.e.*, $G = u_0^k$, the $\hbar^2$ (torus) term reduces to $-\frac{2k+1}{24kx^2}$, as it should.

The second torus term is simply a copy of the result in equation (31), and so will again $x$-integrate twice nicely. The term in braces in the first term can be seen to be the second $x$-derivative of $-\ln(u_0)$, and so:

$$Z_B = -\frac{1}{8}\ln(u_0) + \frac{1}{12}\ln(-u_0') . \tag{43}$$

Finally therefore, combining with (36):

$$Z_{\text{even}} \equiv \frac{1}{2}(Z_A + Z_B) = -\frac{1}{16}\ln\left(-\frac{u_0}{u_0'}\right) , \tag{44}$$

a pleasingly simple result. The leading behaviour of $u_0(x)$ for the $x < 0$ regime is linear: $u_0 = -t_1^{-1}x + \cdots$, and hence result (30) is confirmed. The results are summarized in Table III.

| | $x < 0$ | $x > 0$ |
|---|---|---|
| 0A | $-\frac{1}{16}\ln[(-u_0')^{-\frac{2}{3}}]$ | $-\frac{1}{8}$ |
| 0B | $-\frac{1}{16}\ln[u_0^2(-u_0')^{-\frac{4}{3}}]$ | $0$ |

Table III. The general expressions for the torus partition function in the two perturbative directions for type 0A and 0B theories. For the supersymmetric Virasoro minimal string, function $u_0$ satisfies equation (70), following from the choice of $t_k$ given in equation (8). These formulae hold for more general choices of $t_k$.

## V. OBSERVABLES I: LOOPS AND POINTS

The business of defining the string theories does not end with finding solutions to the string equations. The complete matrix model determines exactly how the orthogonal polynomials, thus found, are to be used to determine observables of the 0A and 0B theories.

### A. Loops: Little and Large

Starting with type 0A, once a solution $u(x)$ is found to the string equation (given appropriate boundary conditions, see below), an efficient way to compute observables in general is through the fact that $u(x)$ supplies a potential for an associated quantum mechanics problem with Hamiltonian $\mathcal{H} = -\hbar^2\partial_x^2 + u(x)$. This object arises as the double scaling limit of the operator representing multiplication of orthogonal polynomials by $\lambda$, which changes their order. Hence, its eigenvalues give information about the energies, which in the scaling limit are labelled $E$. The eigenfunctions $\psi(E,x)$ are indeed the scaled orthogonal polynomials in the limit.

A fundamental observable is the expectation value of finite sized holes in the worldsheets. In the matrix model such holes are made by inserting high powers of the matrix $H$ into the potential. Correlation functions of such operators are naturally computed in the auxiliary quantum mechanics governed by $\mathcal{H}$. For the expectation value of the most basic such "macroscopic loop" operator (of length $\beta$), the computation is as follows:

$$\langle \text{Tr}(e^{-\beta H}) \rangle = \int_{-\infty}^{\mu} \langle x|e^{-\beta\mathcal{H}}|x\rangle \, dx , \tag{45}$$

where the left hand side means an average in the ensemble of matrices $H$, while the right hand side is a computation in the quantum mechanics. (They are *not* to be confused: The $H$'s have discrete spectra while $\mathcal{H}$ has a continuous spectrum - it contains *statistical* information about all possible $H$ spectra.) The number $\mu$ will be further discussed below. It is fixed by comparing the leading part of equation (45) to the leading gravity physics.

The content of the above can be understood at leading order by inserting a complete set of momentum eigenstates $1 = \int dp\, |p\rangle\langle p|$, using that $\langle x|p\rangle = (2\pi\hbar)^{-1}\exp(ipx/\hbar)$, and dropping commutators to leading order, so that

$$\langle \text{Tr}(e^{-\beta H}) \rangle_0 = \frac{1}{2\hbar\sqrt{\pi\beta}} \int_{-\infty}^{\mu} dx\, e^{-\beta u_0(x)} + O(\hbar) \tag{46}$$

$$= \frac{1}{2\hbar\sqrt{\pi}} \int_{-\infty}^{\mu} dx \sum_{l=0}^{\infty} (-1)^l \frac{\beta^{l-\frac{1}{2}}}{l!} u_0^l(x) + O(\hbar) .$$

This is precisely the leading object denoted $Z_0(\beta)$ in Sections I A and I B. Section VI A will discuss how to incorporate the leading behaviour of the string equation into it and directly derive equation (9). For SVMS, comparison to the leading spectral density (7) (the inverse Laplace transform of $Z_0(\beta)$) will fix $\mu$ to be a positive number in loop these calculations. See equation (8).

Going further, the corrections to this amplitude arrange themselves elegantly into:

$$\langle \text{Tr}(e^{-\beta H}) \rangle = \frac{1}{2\hbar\sqrt{\pi}} \int_{-\infty}^{\mu} dx \sum_{l=0}^{\infty} (-1)^l \frac{\beta^{l-\frac{1}{2}}}{l!} R_l[u] + \cdots ,$$
(47)

where the ellipses denote non-perturbative terms, and $R_l[u]$ are the aforementioned $l$th Gel'fand-Dikii polynomials in $u(x)$ and its derivatives[12], normalized such that $u^l$ has coefficient unity.

A macroscopic loop can be thought of as being a sum of an infinite set of "microscopic loop" operators [60], i.e., insertions of point-like operators, denoted here $\mathcal{O}_k$. Picking out the pole in $\beta$ (and stripping away cluttering factors):

$$\begin{aligned} \langle \mathcal{O}_k \rangle &= \frac{c_k\sqrt{\pi}}{\hbar} \frac{(k+1)!}{(-1)^{k+1}} \oint \frac{d\beta}{2\pi i} \beta^{-(k+\frac{3}{2})} \langle \text{Tr}(e^{-\beta H}) \rangle \\ &= \frac{1}{2\hbar^2} \int_{-\infty}^{\mu} dx \int^{x} d\tilde{x} \, c_k R'_{k+1}[u(\tilde{x})] , \end{aligned}$$
(48)

where $\tilde{x}$ is a dummy variable, and $c_k$ is a normalization constant given in footnote 12. What has emerged here is a correspondence between inserting operator $\mathcal{O}_k$ and differentiation with respect to the $t_k$, through the generalized integrable KdV flows:

$$\frac{\partial}{\partial t_k} u(x, \{t_k\}) = c_k R'_{k+1}[u] .$$
(49)

The factors were chosen in the above so that taking two $x$-derivatives on each side and identifying $t_0=x$ recovers (21), the relation between the free energy $F$ and $u(x)$.

Turning to type 0B, the natural integrable flows are given by the "modified" generalized KdV flows, acting on the variable $r(x)$:

$$\frac{\partial}{\partial t_k} r(x, \{t_k\}) = \frac{c_k}{2} K'_{2k}[r] ,$$
(50)

where $t_{2k}$ has simply been written as $t_k$ for short. In fact, the flow equations are linked by the fact that $r$ is related to $u$ by $u(x) = r^2(x) \pm \hbar r'(x)$, and using the identity:

$$K_{2k}[r] = \pm\frac{1}{2} R'_k[r^2 \pm \hbar r'] - r R_k[r^2 \pm \hbar r'] ,$$
(51)

along with the recursion relation for $\widetilde{R}_k = c_k R_k$:

$$\widetilde{R}'_{k+1} = u\widetilde{R}'_k - \frac{\hbar^2}{4} \widetilde{R}'''_k - \frac{1}{2} u' \widetilde{R}_k .$$
(52)

Considering pointlike operators now reveals a key feature, if it is recalled that $r$ comes from having combined the two different $u$ string equations solutions: $r^2 = \frac{1}{2}[u_{\frac{1}{2}} + u_{-\frac{1}{2}}]$. Acting on the 0B free energy (22):

$$\frac{\hbar^2}{c_k} \frac{\partial F}{\partial t_k} = \frac{1}{2} \int_{-\infty}^{\mu} \left[ R_{k+1}[u_{\frac{1}{2}}] + R_{k+1}[u_{-\frac{1}{2}}] \right] .$$
(53)

This means that insertions of pointlike operators in type 0B yields two *separate* sets of KdV flows (since the $u$s are different). This indeed defines two different Schrödinger Hamiltonians, labeled by $s=\pm 1$: $\mathcal{H}_s \equiv -\hbar^2 \partial_x^2 + u_{\frac{s}{2}}$, where $u_{\frac{s}{2}} = r^2 + s\hbar r'$. This shows that the 0B macroscopic loop operator (from which these come by an expansion analogous to (47)) is built by summing two sectors:

$$\langle \text{Tr}(e^{-\beta H}) \rangle = \frac{1}{2} \sum_{s=\pm} \int_{-\infty}^{\mu} \langle x|e^{-\beta \mathcal{H}_s}|x\rangle \, dx .$$
(54)

Note that this is quite different than defining a loop using a single Hamiltonian whose potential is $r^2$, as perhaps might have been the natural guess. That there are two such sectors arises from the fact that the orthogonal polynomials for this problem break into two families[38], moded even and odd, with separate recursion relations and coefficients, which ultimately yield the two $\mathcal{H}_s$.[13]

## B. Perturbing beyond leading order

This is how perturbation theory for loop correlators is developed. A useful picture to have in mind is the following. Recall that the orthogonal polynomials started out being labeled by a discrete index $n$. There are an infinite number of them, but only $N$ of them are used to express matrix model quantities. In the double scaling limit, they are now labeled by a continuous coordinate $x$. The fact that only some of them are used to build the matrix model quantities is the origin of integrating $x$ from $-\infty$ to $\mu$. In an equivalent fermionic many-body language, integrating over this $x$-range is like filling up a Fermi sea to Fermi level $x=\mu$. The physics is then determined by the local behaviour of the Fermi surface, controlled by $u(x)$ and its derivatives at $x=\mu$.

In determining the loop amplitudes (45) (54) for these supersymmetric cases, $\mu$ is in the positive $x$ regime, corrections away from the $u_0 = 0$ behaviour are determined by the positive $x$ expansion of the string equation around that solution, given in expression (20). For the 0A string theory, this was explored, and it was shown how this results in expressions for the "generalized volumes", the

---

[12] A direct way to see this is to note that the diagonal resolvent $\widehat{R}(\sigma, x)$ defined above (38) has an expansion $\widehat{R}(\sigma, x) = \sum_{l=0} c_k(-1)^l \sigma^{-l-\frac{1}{2}} \widehat{R}_l[u]$ where $c_k = \frac{(-1)^l (2l-1)!}{(4)^l l!(l-1)!}$ and an inverse Laplace transform with respect to $\sigma$ gives the result.

[13] This should correlate in the continuum theory to the fact that the FZZT branes in 0B are constructed by combining sectors of opposite charge half RR flux, which nicely correlates here with $\Gamma = \pm\frac{1}{2}$.

natural scattering amplitudes $\widehat{V}_{g,n}(\{P_1\cdots P_n\})$ of the supersymmetric Virasoro minimal string.

Since much of that was done in a normalization that does not match that used here (especially for comparison to the CFT$_2$/3D chiral gravity setup), it is prudent to do a translation, while also unpacking things in a manner that confirms the normalization of the trumpet partition function $Z_{\text{tr}}(\beta, P)$ in equation (4). The key is that, as

made explicit in [19, 20], the volumes $\widehat{V}_{g,1}(P)$ are actually already contained in the expression (47) through the appearance of the Gel'fand-Dikii resolvent $\widehat{R}(x, E)$, and since it satisfies an ODE (38), they can be computed quite efficiently to any desired order.[14]

Following the breadcrumbs left in footnote 12, one can write the 0A loop as:

$$\langle \text{Tr}(\text{e}^{-\beta\mathcal{H}})\rangle \;=\; \int d\sigma\, \text{e}^{\beta\sigma} \int_{-\infty}^{\mu} \langle x| \frac{1}{\mathcal{H}-\sigma}|x\rangle\, dx \quad = \frac{1}{\hbar}\int d\sigma\, \text{e}^{\beta\sigma} \int_{-\infty}^{\mu} \widehat{R}(x,\sigma)\, dx$$

$$= \frac{1}{\hbar}\int d\sigma\, \text{e}^{\beta\sigma} \int_{-\infty}^{\mu} \sum_{g=1}^{\infty} \widehat{R}_g(x,\sigma)dx\, \hbar^{2g} + \cdots \quad = \int \sqrt{2}dz\, \text{e}^{-\beta z^2} \sum_{g=1}^{\infty} \omega_{g,1}(z)\hbar^{2g-1} + \cdots , \qquad (55)$$

where in the penultimate line the perturbative solution to the Gel'fand-Dikii equation (38) is to be used (see comments and explicit form below). Meanwhile, in the last line a change in variables $\sigma = -z^2$ was performed, and the perturbative objects $\omega_{g,1}$ have been defined, and written as Laplace transforms of the $\widehat{V}_{g,1}$:

$$\omega_{g,1} \;\equiv\; \sqrt{2}z \int_{-\infty}^{\mu} \sum_{g=1}^{\infty} \widehat{R}_g(x,z)\, dx = \int_0^{\infty} 2PdP\, \text{e}^{-4\pi Pz}\, \widehat{V}_{g,1}(P) \;. \qquad (56)$$

The $\widehat{R}_g(x,\sigma)$ are defined by recursively expanding the Gel'fand-Dikii equation for general $u(x)$ and then further expanding it by writing $u(x) = u_0(x) + u_2(x)\hbar^2 + u_4(x)\hbar^4 + \cdots$ where the relation between the $u_i(x)$ and the leading solution $u_0(x)$ are to be determined by the string equation (there are only even powers of $\hbar$ in the $x > 0$ regime). The result was derived to fourth order in refs. [19, 20] and keeping the first two orders:

$$\widehat{R}(x,\sigma) \;=\; -\frac{1}{2}\frac{1}{[\sigma]^{1/2}} + \sum_{g=1}^{\infty} \widehat{R}_g(x,\sigma)h^{2g} + \cdots$$

$$= -\frac{1}{2}\frac{1}{[u_0(x)-\sigma]^{1/2}} + \frac{\hbar^2}{64}\left\{ \frac{16u_2(x)}{[u_0(x)-\sigma]^{3/2}} + \frac{4u_0''(x)}{[u_0(x)-\sigma]^{5/2}} - \frac{5(u_0'(x))^2}{[u_0(x)-\sigma]^{7/2}} \right\}$$

$$+ \frac{\hbar^4}{4096}\left\{ \frac{1024u_4(x)}{[u_0(x)-\sigma]^{3/2}} - \frac{256[3u_2(x)^2 - u_2''(x)]}{[u_0(x)-\sigma]^{5/2}} + \frac{64[u_0^{(4)}(x) - 10u_2(x)u_0''(x) - 10u_0'(x)u_2'(x)]}{[u_0(x)-\sigma]^{7/2}} \right.$$

$$\left. - \frac{16[28u_0^{(3)}(x)u_0'(x) + 21u_0''(x)^2 - 70u_2(x)u_0(x)^2]}{[u_0(x)-\sigma]^{9/2}} + \frac{1704u_0'(x)^2 u_0''(x)}{[u_0(x)-\sigma]^{11/2}} - \frac{1155u_0'(x)^4}{[u_0(x)-\sigma]^{13/2}} \right\} + \cdots , \quad (57)$$

and the opposite sign choice to that made earlier in (39) has been made.

It is remarkable (observed in ref. [19], and explored further in refs.[59, 61]) that at any order in $\hbar$, upon using the string equation to relate the higher order $u_i(x)$ to $u_0(x)$ and its derivatives, the quantity $\widehat{R}_g(x,\sigma)$ is in fact a total derivative. Although this has not yet been proven[15], it *must* be so, since (guided by the ordinary case of Weil-Peterson volumes to which this problem reduces) the $\omega_{g,n}(\{z_i\})$ (which result from integrating them) should be polynomials in inverse powers of $z_i$, since the $\widehat{V}_{g,n}(\{P_i\})$ are polynomials in the $\{P_i\}$. That the $\widehat{R}_g(x,\sigma)$ are total derivatives also makes sense since then the integrals in the first line of (56) then depend just on the behaviour at the boundary $x=\mu$, fitting nicely with the Fermi sea picture given at the beginning of this section.

---

[14] The ODE also also contains non-perturbative information as well, which translates into valuable non-perturbative data about the volumes, which could be interesting to explore.

[15] Since both sets of expansions involve properties of Gel'fand-Dikii polynomials, a proof seems likely to follow from the underlying KdV integrability.

Before exploring examples, putting (56) into (55) and doing the $z$-integral gives:

$$\langle\mathrm{Tr}(\mathrm{e}^{-\beta\mathcal{H}})\rangle = \langle\mathrm{Tr}(\mathrm{e}^{-\beta\mathcal{H}})\rangle_0 + \sqrt{\frac{2\pi}{\beta}}\int_0^\infty 2PdPe^{-\frac{4\pi^2 P^2}{\beta}}\sum_{g=1}^\infty \widehat{V}_{g,n}(P)\hbar^{2g-1} + \cdots$$

and the last term is:

$$\sum_{g=1}^\infty Z_g(\beta) = \sum_{g=1}^\infty \int_0^\infty 2PdPZ_{\mathrm{tr}}(\beta,P)\widehat{V}_{g,n}(P)\hbar^{2g-1} + \cdots, \quad \text{with } Z_{\mathrm{tr}}(\beta,P) = \sqrt{\frac{2\pi}{\beta}}\mathrm{e}^{-\frac{4\pi^2 P^2}{\beta}}, \tag{58}$$

matching (4), which like the leading loop expectation value (9), was shown in Section I B to have a nice CFT$_2$ interpretation, presumably describing a dual 3D chiral supergravity system.

It doesn't hurt at this point to be more explicit. Solving the string equation for $u(x)$ to fourth order gives:

$$u(x) = 0 + \left(\Gamma^2 - \frac{1}{4}\right)\frac{\hbar^2}{x^2} - 2t_1\left(\Gamma^2 - \frac{1}{4}\right)\left(\Gamma^2 - \frac{9}{4}\right)\frac{\hbar^4}{x^5} + \cdots \tag{59}$$

and so (the integrals in this $x > 0$ regime are elementary compared to the $x < 0$ regime where the total derivative result is essential):

$$\omega_{1,1} = -\frac{\sqrt{2}}{4}\left(\Gamma^2 - \frac{1}{4}\right)\frac{1}{z^2}\frac{1}{\mu}, \quad \omega_{2,1} = \frac{\sqrt{2}}{16}\left(\Gamma^2 - \frac{1}{4}\right)\left(\Gamma^2 - \frac{9}{4}\right)\left(\frac{2t_1}{z^2\mu^4} + \frac{1}{z^4\mu^3}\right), \tag{60}$$

giving:

$$\widehat{V}_{1,1}^{(b)}(P) = -\frac{(4\pi)^2}{2}\left(\Gamma^2 - \frac{1}{4}\right)\left(\frac{\sqrt{2}}{4\mu}\right), \quad \widehat{V}_{2,1}^{(b)}(P) = \frac{(4\pi)^2}{2}\frac{\sqrt{2}}{96\mu^3}\left(\Gamma^2 - \frac{1}{4}\right)\left(\Gamma^2 - \frac{9}{4}\right)\left((4\pi P)^2 + \frac{12t_1}{\mu}\right). \tag{61}$$

(Ref. [20] also calculated $\widehat{V}_{3,1}$. That can be translated into the conventions here by sending $P_1$(there) to $4\pi P$ and multiplying overall by $(4\pi)^2/2$.) Into these expressions can be substituted the values (from (8)) $\mu = 4\sqrt{2}\pi$ and:

$$t_1 = 2\sqrt{2}\pi^3(Q^2 + \widehat{Q}^2) = 4\sqrt{2}\pi^3\left(b^2 + \frac{1}{b^2}\right), \tag{62}$$

resulting in the nicer expressions:

$$\widehat{V}_{1,1}^{(b)}(P) = -\frac{\pi}{2}\left(\Gamma^2 - \frac{1}{4}\right), \quad \widehat{V}_{2,1}^{(b)}(P) = \frac{\pi}{96}\left(\Gamma^2 - \frac{1}{4}\right)\left(\Gamma^2 - \frac{9}{4}\right)\left(P^2 + \frac{3}{4}\left(b^2 + \frac{1}{b^2}\right)\right). \tag{63}$$

For $\Gamma = 0$, the 0A case in the (1,2) ensemble, $\widehat{V}_{1,1}$ matches that of the 0A super JT case [13]), up to a factor. Meanwhile, writing in terms of the central charge:

$$\widehat{V}_{2,1}(P) = \frac{3\pi}{512}\left(P^2 + \frac{1}{4}\left(c - \frac{15}{2}\right)\right) \tag{64}$$

which (on sending $b^2 \to 0$ and $P \to 0$ holding $bP \to \ell/(4\pi)$ fixed) reduces to the JT supergravity result [13] for the relevant super Weil-Peterson volume for geodesic boundary length $\ell$ (again up to a factor). More interesting for the case in hand is the form, which compares well to the ordinary VMS quantum volume computed in ref. [16]: $\widehat{V}_{1,1} = \frac{1}{24}\left(P^2 + \frac{c-13}{24}\right)$. The Liouville $c$ minus half the total central charge, with an overall factor, accompanies the $P^2$, in both cases. It would be interesting to see if there is a simple interpetation of this in continuum approaches.

For the cases $\Gamma = \pm\frac{1}{2}$ the volumes $\widehat{V}_{g,1}(P)$ vanish, because $u(x)$ vanishes to all orders in perturbation theory. It should be clear now that this structure (and hence the prediction) carries over to the 0B string theory, since its loop observables are constructed, in an analogous way to the procedures above, precisely from these same $\Gamma = \pm\frac{1}{2}$ building blocks.

More generally, $u(x)$ and its derivatives for $x > 0$ control $n$-loop scattering amplitudes for 0A SVMS as well (or $r^2(x) = \frac{1}{2}[u_{+\frac{1}{2}} + u_{-\frac{1}{2}}]$ and its derivatives, in the direct 0B language). There will generically be non-zero results for 0A (they won't be explored here), but for $\Gamma = \pm\frac{1}{2}$ they will again all vanish for $n \geq 3$, and the same is true for 0B SVMS. One way to see that [35] is through the fact that loops can be deconstructed into an expansion in terms of

insertions of the "pointlike" operators as reviewed in Section V A. Such insertions are given to all orders by derivatives of $u(x)$ $(r(x))$ through the KdV (mKdV) flows. So exact vanishing (to all orders) of those functions in the $\Gamma=\pm\frac{1}{2}$ and 0B cases results in the vanishing of all higher-point loops to all genus. Again, this would be very interesting to see emerge using other approaches, and constitutes a robust prediction.

A special exception is the two-loop case. At leading order this is a cylinder diagram, with two asymptotic boundaries of lengths $\beta_1$ and $\beta_2$. The amplitude is constructed by simply sewing together two copies of the trumpet [12, 16], (see equation (58)), joining them at the geodesic boundary by integrating over the length $P$ using the same $2PdP$ measure, and the result is the well-known form universal [62–65] to a wide class of random matrix models:

$$\langle \mathrm{Tr}(\mathrm{e}^{-\beta_1 H})\mathrm{Tr}(\mathrm{e}^{-\beta_2 H})\rangle_0 = \frac{\sqrt{\beta_1\beta_2}}{2\pi(\beta_1+\beta_2)} \ . \qquad (65)$$

Higher genus corrections to this are generated again by derivatives of $u(x)$, and so do not exist when those vanish.

## VI.  OBSERVABLES, II: SPECTRAL DENSITIES

The single loop amplitude also directly yields information about the spectral density $\rho(E)$ of the matrix model. Knowing the spectrum of eigenfunctions of $\mathcal{H}$ (or the two $\mathcal{H}_s$) is key. For 0A, inserting a complete set of energy eigenstates, one can write (45) as:

$$\langle \mathrm{Tr}(\mathrm{e}^{-\beta H})\rangle = \int_0^\infty \left\{ \int_{-\infty}^\mu \langle x|E\rangle\langle E|x\rangle \, dx \right\} \mathrm{e}^{-\beta E} dE$$

$$= \int_0^\infty \rho(E)\mathrm{e}^{-\beta E} dE \ , \qquad (66)$$

where the wavefunctions are $\psi(E,x)\equiv\langle E|x\rangle$, and the combination:

$$\rho(E)=\int_{-\infty}^\mu \psi(E,x)^2 dx \qquad (67)$$

is the full non-perturbative spectral density. A similar set of manipulations for 0B gives:

$$\langle \mathrm{Tr}(\mathrm{e}^{-\beta H})\rangle = \sum_{s=\pm} \int_0^\infty \rho_s(E)\mathrm{e}^{-\beta E} dE \ , \qquad (68)$$

where $\rho_s(E)$ is the density from each sector.

Recalling that $H=Q^2$, it is the Hermitian matrix $Q$ that has the double well, so it is natural to work with the spectrum of $q$ eigenvalues, which runs from $-\infty$ to $+\infty$, and density $\rho(q)=\frac{q}{2}[\rho_-(q^2) + \rho_+(q^2)]$ .

### A.  Leading order

First the leading order solution of the string equation must be found. Setting $\hbar=0$ in the string equation (14)

$u_0(x)$, the leading part (in an $\hbar$ expansion) of $u(x)$, solves $\sum_{k=1}^\infty t_k u_0^k(x)+x=0$, for $x < 0$ and $u_0=0$ for $x>0$.

The machinery for using this piecewise solution to yield the particular properties needed for supersymmetric cases was developed in refs. [26, 34, 66], and details can be found there. It can be done either in terms of leading spectral densities $\rho_0(E)$, or in terms of the leading amplitude for a loop of length $\beta$, $Z_0(\beta) \equiv \langle \mathrm{Tr}(\mathrm{e}^{-\beta H})\rangle_0$. They are Laplace transform pairs.

Start with the former. Briefly, after taking the leading Wentzel-Kramers-Brillouin (WKB) form of the wavefunctions (with a choice of normalization) the spectral density above can be written, for 0A as:

$$\rho_0^{(b)}(E) = \frac{1}{2\pi\hbar} \int_{-\infty}^\mu \frac{\Theta(E-u_0(x))dx}{\sqrt{E-u_0(x)}}$$

$$= \frac{1}{2\pi\hbar} \int_0^E \frac{f(u_0)du_0}{\sqrt{E-u_0}} + \frac{\mu}{2\pi\hbar\sqrt{E}} \ , \qquad (69)$$

where $-f(u_0)=\partial x/\partial u_0$ is the Jacobian in going from $x$ to $u_0$. and $u_0(x)$ is the leading part (in an $\hbar$ expansion) of the function $u(x)$, with $u_0(\mu)=0$.

The first piece in the last line of (69) results from the integral over $x$ from $-\infty$ to 0, and used the fact that $u_0(0)=0$. It determines most of the bulk of the leading spectral density. The second term there comes from the integral from $x=0$ to $x=\mu$, a region where $u_0(x)=0$. It produces the key leading $E^{-\frac{1}{2}}$ behaviour of this class of models.

The fact that $f=\sum_{k=1}^\infty k u_0^{k-1}$ yields a series of $u_0$ integrals that result in $\rho_0(E)=\sum_{k=0}^\infty D_k E^{k-\frac{1}{2}}$, for numbers $D_k$ that depend on the $t_k$ for $k \geq 1$. ($D_0$ determines $t_0\equiv\mu$.) Comparing this series to the proposed leading spectral density (7), also expressed as a series in $E$, neatly determines the $t_k$. This was done in ref. [20] and the result is expressions (8). This yielded an interesting form for the leading string equation in the $x<0$ regime:

$$2\sqrt{2}\pi \left[ I_0(2\pi Q\sqrt{u_0}) + I_0(2\pi\widehat{Q}\sqrt{u_0}) \right] + x = 0 \ . \quad (70)$$

Here $I_0(y)$ is the modified Bessel function in $y$ of order 0, and this leading string equation was analyzed in ref. [20], as well as perturbative and non-perturbative solutions of the full string equation obtained by using this (with $u_0=0$ for $x>0$) as the leading solution.

The alternative route, matching $Z_0(\beta)$ instead, is done by directly solving the integral in equation (46), by changing from an $x$-integral to a $u_0$ integral. Jacobian $-f(u_0)$ given above again gives a series of elementary integrals.

It is now straightforward to do all this for the 0B case. At leading order, the potentials $u_{\frac{s}{2}}$ are simply $r_0^2$, where $r_0(x)$ is the leading solution to the string equation (15) with $\hbar$ set to zero. Since $K_{2k}\to r_0^{2k+1}$, an overall factor of $r_0$ can be pulled out, and so the string equation also breaks into two sectors: $\sum_{k=1}^\infty t_k r_0^{2k}(x)+x=0$, for $x<0$ and $r_0(x)=0$ for $x>0$. Meanwhile the WKB form of

the wavefunctions yields the same form for the leading spectral density as before, with $r_0^2$ replacing $u_0$ in equation (69). The key point then is that *exactly* the same form of $t_k$ as for the type 0A case will work here to determine the spectral density for the 0B case. That this had to be true will be discussed from the string worldsheet perspective in Section IV.

### B. Non-perturbative physics

The next step is to solve equation (14) for $u(x)$ for the two cases $\Gamma=\pm\frac{1}{2}$. The result is shown in figure 7, for the example of $\hbar=1$, with a truncation to $k=7$ for $b=1$ for example.[16] The problem of computing the spectrum of wavefunctions and energy can be done for each case (in ref. [20] the $\Gamma=0,+\frac{1}{2}$ cases were presented) and used to compute the full spectral densities of $H$ for each model, according to (67). The results are added according to equation (54), and all three are presented (as a function of $E$) in figure 8. Finally, this can be converted into the spectrum for the operator $Q$, which is displayed in figure 9.

Notable is the fact that the peaks and troughs of the separate components (almost) cancel each other out, resulting in a non-perturbative spectrum that has very suppressed undulations. In fact, as a result, the leading spectral curve $\rho_0(q)$ is (on the scale shown) indistinguishable from the complete non-perturbative solution $\rho(q)$!

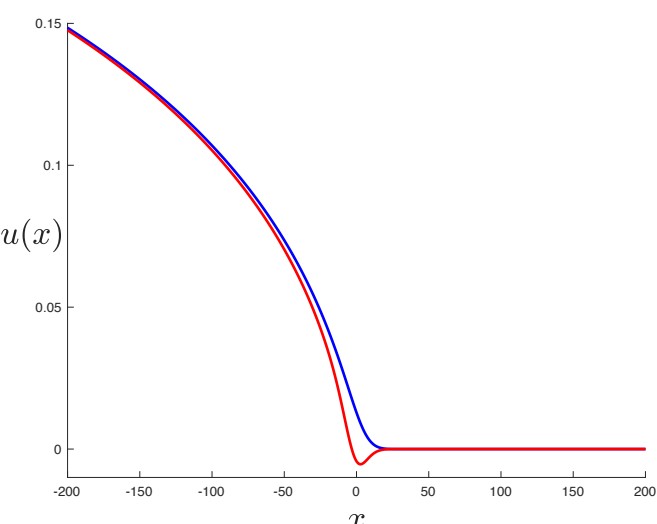

Figure 7. The functions/potentials $u_0(x)$ for the cases $\Gamma=-\frac{1}{2}$ (red, lower) and $\Gamma=+\frac{1}{2}$ (blue, upper) for the $b=1$ case.

---

[16] The process of truncating the string equation at high enough $k$ in order to get (well controlled) numerical results was described in refs. [34, 66].

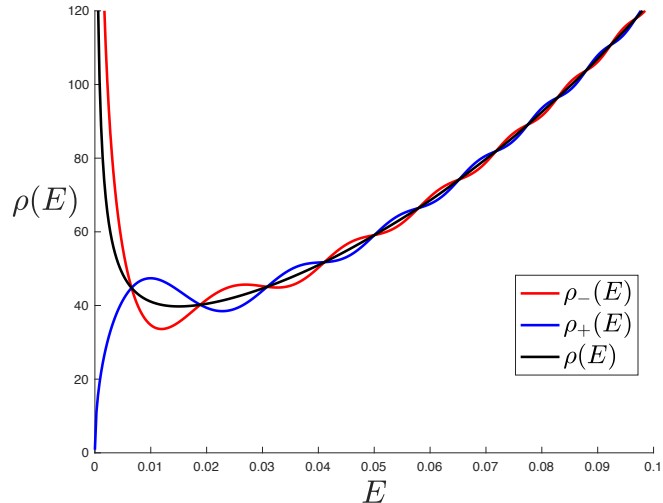

Figure 8. The densities of eigenvalues for $H$, for the special 0A models: $\Gamma=+\frac{1}{2}$ (blue, passing through 0) and $\Gamma=-\frac{1}{2}$ (red). Their average is shown in black. This is for the $b=1$ case.

In fact, this matches expectations for a merged two-cut system. See footnote 7 where some details of the intuition/expectation are discussed.

### C. A note on dual loops

As a final note, it is worth remarking upon an interesting feature of the 0A–0B duality of $k=1$. The closed

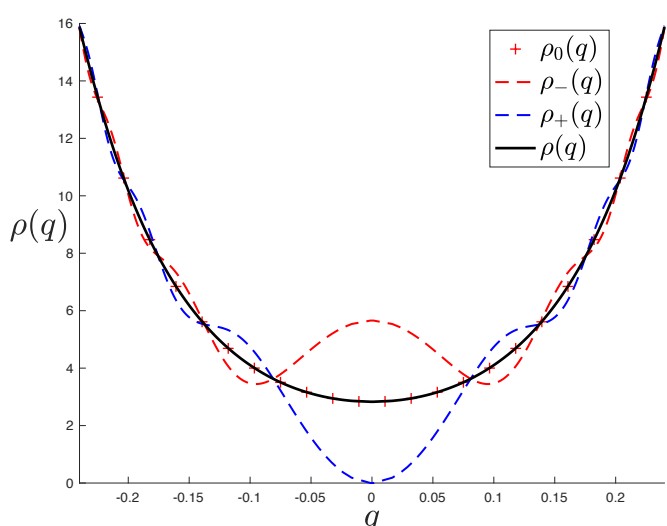

Figure 9. The density of eigenvalues for $Q$, for the type 0B model, is shown in black (central line). It is the average of contributions from the $\Gamma=+\frac{1}{2}$ sector (blue, passing through 0) and the $\Gamma=-\frac{1}{2}$ sector (red). The total result suppresses the non-perturbative peaks, resulting in something very close to the classical result, a few values of which are shown with red crosses. This is for the $b=1$ case, but other values have similar behaviour.

string sectors were seen to map into each other, up to an $x$ additive term between $u(x)$ and $r(x)^2$ that led amounts to a (non-universal) difference in the contribution at the sphere level. This might lead the unwary to the conclusion that the complete random matrix models are identical. But clearly the spectral densities can be computed are quite different, given what was seen above. There is no paradox, however. The lesson is that since there are two separate closed string regimes for each theory (one for each of the $\pm x$ directions), there are two classes of open string sector (FZZT-brane, or loop operator) that can be naturally introduced into each theory [67–69]. In each sector, the relevant loop length $\ell$ is the Laplace-transform pair of an energy variable, but it is only for one of the sectors in each model ($+x$) that this energy $E$ should be identified with the double-scaled matrix model eigenvalues. Since $\pm x$ are exchanged under the duality exchanging 0A and 0B, the spectral densities of the random matrix models they are contained within will therefore *not* be the same. (In short, sharing the same string equations, Painlevé II with zero constant, does not mean the matrix models are the same.) (However this does introduce the interesting notion of a natural "dual" eigenvalue distribution for some classes of random matrix models in general, which is perhaps worth exploring.)

## VII. CONCLUSION

This point in the paper feels more like a beginning than an ending, since there are many immediately interesting directions to explore now that the task of introducing the theories and the framework with which to study them is complete. Many aspects of the results uncovered in the paper are discussed in the summary presented in Section I C, and so it seems unnecessary to repeat them here.

The supersymmetric Virasoro minimal string (SVMS) theories (with type 0A and type 0B variants) discussed and presented here, generalize the ordinary VMS pre-

sented in ref. [16]. A key point is that they are fully non-perturbatively well defined, and constitute a continuously infinite pair of families (by varying $b$). They also contain $\mathcal{N}=1$ supersymmetric JT gravity in the $b\to0$ classical limit. Hence, as already remarked when the 0A case was defined in ref. [20], these properties make them especially sharp tools for studying quantum gravity, holography and string theory.

Indeed, as observed in this paper, their properties strongly suggest a description in terms of a $\mathrm{SCFT}_2$ that in turn has a 3D chiral supergravity dual, where presumably many of the properties of supersymmetric Liouville (spacelike and timelike) naturally have a home, in analogy with what happened for the ordinary Virasoro minimal string in ref. [16]. Constructing that description (and the closely associated intersection theory framework) seems like an urgent avenue to pursue. It very much seems that the properties of the 0A and 0B descriptions as random matrix models set out here can give guidance in this endeavour, as well as to other continuum approaches (see *e.g.*, recent work in refs.[21, 22]).

It is worth remarking in closing that a clear and natural generalization of these SVMS models to models with higher supersymmetry seems highly possible. Their random matrix model constructions using the multicritical approach done here should be relatively straightforward, given recent successes [36, 70, 71] in applying such tools to $\mathcal{N}=2$, $\mathcal{N}=3$, and $\mathcal{N}=4$ JT supergravity, which have random matrix model descriptions [72, 73].

### ACKNOWLEDGMENTS

CVJ thanks the US Department of Energy for support (under award #DE-SC 0011687), and Victor Rodriguez and Maciej Kolanowski for remarks. The final preparation of this manuscript was completed at the Aspen Center for Physics, which is supported by National Science Foundation grant PHY-2210452. CVJ thanks Amelia for her support and patience.

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
