# Peer review of "Further aspects of Supersymmetric Virasoro Minimal Strings"

_SciPost Physics_

## Round 1 · Referee Report · Anonymous (Referee 1) · 2025-11-14

Report

This paper proposes and studies a matrix model of a supersymmetric version of the Virasoro Minimal String. The study of the proposed matrix model is generally impressive. I appreciated section IB, section IIB and section III. I furthermore found Figures 8 and 9 useful and nice. Equation (63) is a nice target for worldsheet calculations, as is the vanishing of higher genus amplitudes for Gamma squared equal to a quarter. However the word `target’ is maybe a bit too strong, which brings me to my point of critique.

The author claims that this matrix model is the definition of SVMS, and that worldsheet theories or other techniques should reproduce this. I find this a bit misleading to the readers. The author is making a suggestion that dualities between matrix models and string models are obvious. However, not every string theory is a matrix model, and not every matrix model is a string theory. For instance, deformations of JT gravity do not obviously have a reasonable worldsheet CFT description, so it is difficult to call these string theories. String theory is defined as a worldsheet CFT. It may be that SVMS coincides with the matrix models that have been studied. But it is also possible that it is completely different. The title is perhaps a bit strong, suggesting that what he is doing is certainly related with SVMS. This is in my opinion not necessarily guaranteed. In the conclusion, the author says that this is a sharp tool to study quantum gravity and string theory. This would only be true of SVMS at all genus matches this matrix model, which has to be shown, and is not obviously true.
In conclusion, the work is in my opinion very impressive and a valuable addition to the matrix model literature. However, the relation with a supersymmetric version of VMS needs more clarification. For the paper to be acceptable for publication, the author should change some of the wording.
To be clear, I do believe the paper provides useful and suggestive formulas that can give guidance in developing the worldsheet theory (as he says in the conclusion).

Requested changes

Change wording.

Recommendation

Ask for minor revision

  • validity: top
  • significance: good
  • originality: good
  • clarity: top
  • formatting: excellent
  • grammar: perfect

Author:  Clifford Johnson  on 2025-12-08  [id 6122]

(in reply to Report 1 on 2025-11-14)
Category:
remark
reply to objection

I thank Referee #1 for taking the time to review the paper, and thank them for their positive remarks.

Puzzlingly, three times as much space in their report was given over to why I'm apparently not allowed to call it a supersymmetric virasoro minimal string, so I'd like to address this, using the facts of the matter as well as the long-existing standards of terminology use in our field. The points:

(1) The original authors of the VMS showed that it had three independent constructions: (A) via worldsheet Liouville CFT, (B) via chiral 3D gravity (and/or intersection theory) computations, and (C) as a random matrix model whose leading spectral density is the "cardy" density of states. In fact, it is in (C) that most computations of observables is most easily done. Notably, the density of states formula has a natural interpretation in terms of a "boundary" CFT seen by the chiral gravity setup.

(2) In the paper (and the one to which it is a sequel) I started with a very natural generalization, building matrix models based on the N=1 supersymmetric version of the density of states. This seems to be already a good reason to call it a supersymmetric VMS, because it is supersymetrizing the VMS. In fact no other supersymmetric generaliztions had been constructed and so it is not like there can be any confusion. So being told that I can't say it is a supersymmetric VMS is puzzling.

(3) But I actually go further and show how to describe the matrix model I constructed as a combination of building blocks that are *known* to be string theories, so the claim that this might not be a string theory is even more puzzling!

(4) What I suggest in the paper is that it would be nice to independently derive this string theory using worldsheet CFT methods. This is not an unreasonable thing to suggest as an interesting piece of future work. Nowhere in my paper do I say/suggest that it is obvious, as the referee claims. Moreover, following in the spirit of the original VMS paper, the matrix model is likely to be a very useful laboratory for computing things that might be hard to pin down using the CFT or other approaches... a concrete set of results to aim for. Why is it not ok to say that?

(5) I also note that the referee keeps referring to "the SVMS" that my work may or may not connect with, as though there is some other definition in the literature. I find this interesting, to say the least. The fact is, when I defined this class of matrix models and presented it (in early 2024) as a natural supersymmetric generalization of VMS, there was not a single paper out there presenting an alternative definition. So again I ask what confusion could there be? There was no other SVMS to compare to yet. Only earlier this spring/summer did two papers appear (which I cited/lauded in the current work) doing some of the difficult steps that will be needed to construct a SVMS via CFT methods. They fall short (by their own admission) of defining the SVMS using only CFT methods, and find that they need to appeal to matrix model intuition as a possible guide for how to proceed, interestingly.

(6) The referee also makes the claim "String theory is defined as a worldsheet CFT" in support of their thesis that I'm not allowed to say this is a string theory. Well, I respectfully point out that this is simply wrong. We teach those learning string theory for the first time that the worldsheet CFT definitions that emerge for simple string theories (perturbatively, for the right kinds of field content) is a happy circumstance that need not occur in general. In fact, there are numerous well known string theories in the literature that do not have world sheet CFT definitions, or that started out without such formulations initially. Are all the papers that present results for them wrong in calling them string theories? Should they call them "proposed string theories", "would-be string theories", oor some other term, until the world-sheet CFT definition is found?

In summary, I respectfully decline the suggestion to unnecessarily re-title the paper and re-write the contents to somehow avoid calling what is undisputably a supersymmetric version of the virasoro minimal string by the name "supersymmetric virasoro minimal string".

It seems to me that such a lengthy semantic exercise is really not a good use of anybody's time.

---

## Round 1 · Referee Report · Anonymous (Referee 2) · 2025-12-7

Report

Dear Editor,

The present article continues the authors' previous studies of double-scaled matrix models equivalent to the supersymmetric version of the Virasoro minimal string. Minimal strings, and, equivalently, matrix models, have of course enjoyed an enormous amount of recent attention on account of the insights they have provided for the physics of black holes (particularly at low temperature) and gravitational effective field theory (defined through a path integral) more generally. The Virasoro minimal string is an example of the success of this recent trend, with a fairly simple worldsheet model (timelike Liouville and spacelike Liouville at $c_{\rm tot}=26$ coupled to the usual worldsheet gravity) being equivalent not merely to a matrix model but perhaps also to a model of pure three-dimensional gravity, or at least a chiral version thereof.

The author was the first to develop a supersymmetric version of the VMS, and since then it has been the subject of several researchers' work. In fact, there is not just one version, but two, what are referred to as 0A and 0B in this manuscript. The present manuscript has several goals, fleshing out various aspects of both models. Borrowing from technology developed in part by the author in the 90's, a particularly novel theme running through this paper is to understand the 0B matrix model in detail, both in its genus expansion and non-perturbatively. As with the VMS, there is some reason to think (see especially the discussion at the end of the Introduction) that there is a connection with 3d gravity, on account of the leading density of states (for 0A) being the primary-counting density of states one infers from the $S$-transform of the chiral NS-R vacuum character of a SCFT.

The paper was clear, timely, interesting, although it left me with some questions whose answers may well be in the manuscript. When the dust settles, what is the leading density of states of the 0B model? Is there a natural 3d interpretation of it? Or, for that matter, a relation between 0A and 0B on the basis of discrete symmetries and their gauging? And, going back to the beginning, I wondered if the manuscript might benefit from a recap of the main tool used in the manuscript -- mapping one string equation to another and thereby defining the 0B model -- from the worldsheet point of view, since the method, while clear enough on its own terms, is not (as far as I know) widely appreciated.

These questions are more for my own personal understanding and I leave it to the authors' discretion whether they should be answered here or via changes in the manuscript.

Recommendation

Publish (easily meets expectations and criteria for this Journal; among top 50%)

  • validity: top
  • significance: -
  • originality: -
  • clarity: -
  • formatting: -
  • grammar: -

Author:  Clifford Johnson  on 2025-12-09  [id 6123]

(in reply to Report 2 on 2025-12-07)
Category:
answer to question

I thank Referee #2 for taking the time to review the paper and provide a report, for their positive remarks, and their engagement with the material in the form of interesting questions, and a suggestion.

To the questions:

The leading 0B spectral density is discussed in section VI A "leading order", where it is observed that it is the same as that of 0A. It is in the corrections that the two theories will differ, as well as in multi-point correlators. This is all in the paper around that area. This is very much in accord with the known structures for 0A an 0B JT supergravity constructions, (to which I refer, starting with Stanford and Witten 2019 and my own constructions that built on that) and so I felt it was not needed to spell it out too much. I worry sometimes that in trying to be clear in my papers on this subject, I sometimes spell out too much, make the paper too long, and give the impression that I'm writing a review, so I tried to be careful here and instead pointed to the literature. I'm sorry if it was missed in the reading. I'll look again to make sure it is not buried too much.

Following from that, the (putative) 3D chiral (super)gravity suggestion I make for 0A interpretation (through the CFT) is then the same suggestion for 0B. Thanks for asking.

For the suggested recap/review about mapping between different string equations can map string theories to each other, I was again concerned about including too much review so as to obscure the new results. But in fact, in section IV A and B I do a decent amount of review of the basics going back to old work of mine (with collaborators) in 1992 but also connecting to the work of KMS in 2003 who rediscover and extend such observations (making the 0A/0B interpretation clear). But then, the section IV B+C (and later) goes on to derive new results from those maps that then explain both the CFT 0A/0B relation and the modern 0A/0B JT connections.

Yes, it is rather like a discrete map, as emphasized in section IV, equations (25)-(27), showing explicitly how a sum of string equation expansions with one choice of signs gives 0A while with antoher gives 0B. (It is strikingly reminiscent of worldsheet CFT methods, as indeed I point out, but this is fully non-perturbative.)

As stated in the paper, I expect that there could be a nice role for these relations between 0A and 0B emerging naturally in the chiral 3D gravity picture. I think this is exciting to pursue.

I appreciate that the suggestion was left as an optional action. Nevertheless I will carefully look over the paper again to see if I have the right balance of review/exposition (vs guiding the reader to the literature).

---

## Editorial Decision

awaiting_resubmission